# A molecular switch in sulfur metabolism to reduce arsenic and enrich selenium in rice grain

Sheng-Kai Sun [1], Xuejie Xu[1], Zhong Tang[1], Zhu Tang[1], Xin-Yuan Huang [1], Markus Wirtz [2], Rüdiger Hell[2] & Fang-Jie Zhao [1✉]

Rice grains typically contain high levels of toxic arsenic but low levels of the essential micronutrient selenium. Anthropogenic arsenic contamination of paddy soils exacerbates arsenic toxicity in rice crops resulting in substantial yield losses. Here, we report the identification of the gain-of-function *arsenite tolerant 1* (*astol1*) mutant of rice that benefits from enhanced sulfur and selenium assimilation, arsenic tolerance, and decreased arsenic accumulation in grains. The *astol1* mutation promotes the physical interaction of the chloroplast-localized *O*-acetylserine (thiol) lyase protein with its interaction partner serine-acetyltransferase in the cysteine synthase complex. Activation of the serine-acetyltransferase in this complex promotes the uptake of sulfate and selenium and enhances the production of cysteine, glutathione, and phytochelatins, resulting in increased tolerance and decreased translocation of arsenic to grains. Our findings uncover the pivotal sensing-function of the cysteine synthase complex in plastids for optimizing stress resilience and grain quality by regulating a fundamental macronutrient assimilation pathway.

[1] State Key Laboratory of Crop Genetics and Germplasm Enhancement, College of Resources and Environmental Sciences, Nanjing Agricultural University, Nanjing 210095, China. [2] Centre for Organismal Studies (COS), Heidelberg University, 69120 Heidelberg, Germany. ✉email: Fangjie.Zhao@njau.edu.cn

Rice is the dominant source of the carcinogenic arsenic (As), accounting for around 60% of the dietary intake of inorganic As for some Asian populations[1–4]. Widespread contamination of paddy soils with As due to mining activities and irrigation of As-laden groundwater not only increases As accumulation in rice grain[5,6], but also results in As toxicity in rice plants and substantial yield losses[7,8]. Worryingly, future climate changes are likely to exacerbate both the problems of As accumulation in rice grain and yield loss induced by As toxicity[9]. It is therefore important to develop rice crops that are more resistant to As and accumulate less As in the grain.

A key mechanism of As detoxification in plants is via the complexation of arsenite [As(III)] by thiol compounds, such as glutathione (GSH) and phytochelatins (PCs), and subsequent sequestration into the vacuoles[10–14]. This mechanism also reduces As translocation to rice grain[12,15–17]. The metal(loid) stress-induced production of GSH and PCs is limited by the availability of its precursor cysteine (Cys) in plants[18–20]. Cys is synthesized in all subcellular compartments with own protein biogenesis by a two-step pathway: the formation of O-acetylserine (OAS) from serine and acetyl CoA catalyzed by serine acetyltransferase (SAT) and the condensation of OAS and sulfide to Cys catalyzed by O-acetylserine(thiol)lyase (OAS-TL)[21,22]. This two-step pathway is conserved in bacteria, alga, and higher plants, and serves as a checkpoint for the integration of sulfur, carbon, and nitrogen. The enzymatic functions of three isoforms of OAS-TL (OAS-TL A, B, and C in the cytosol, plastids, and mitochondria, respectively) in Arabidopsis thaliana and a cytosolic OAS-TL A in rice have been characterized[23–27]. Knockout of each of the three isoforms of OAS-TL in Arabidopsis did not affect growth, because the substrates for Cys biosynthesis can be exchanged efficiently between the cytosol and the organelles[25]. SAT and OAS-TL form the hetero-oligomeric cysteine synthase complex (CSC), and SAT is active only within this complex[21,22]. The product of SAT, OAS, has a dissociation effect on the CSC[22,28,29], thus providing a feedback inhibition on SAT activity. Synthesis of OAS by SAT is also the rate-limiting step of Cys biosynthesis, whereas the total activity of OAS-TL is in excess of that required for Cys biosynthesis[22]. The association of the CSC stimulates the SAT activity and thus provides a mechanism for metabolic regulation of Cys biosynthesis by sensing the supply of the carbon/nitrogen-containing precursor OAS and the sulfur precursor sulfide.

In this study, we isolated a gain-of-function arsenite tolerant 1 (astol1) rice mutant. We show that ASTOL1 encodes the chloroplast-localized OAS-TL. Surprisingly, the astol1 gain-of-function mutation (S189N) inactivates the catalytic activity of OAS-TL by inhibiting OAS binding. We provide direct evidence that the inactive ASTOL1 protein forms a non-dissociable CSC, which permanently stimulates endogenous SAT activity explaining the dominant inheritance of the astol1 mutation. Uncoupling of the OAS-sensing function of the plastidic CSC from its regulatory impact on SAT resulted in (1) enhanced S uptake and assimilation, (2) increased tolerance to As and (3) decreased As accumulation in rice grain. Moreover, the OsASTOL1$^{S189N}$ mutation significantly enhanced the rice grain content of selenium (Se), an essential micronutrient for which up to a billion people worldwide have insufficient dietary intake[30,31]. Besides the potential biotechnological applications, our findings uncover the pivotal sensing-function of the plastid CSC in the regulation of S and Se metabolism and tolerance to metalloid toxicity.

## Results

### Screening and characterization of the astol1 mutant. We screened more than 4000 individuals derived from an EMS-mutagenized library of the rice cv. Kasalath for arsenite [As(III)] tolerance using a root elongation assay at 20 μM As(III) (approximately the half-inhibitory concentration, Supplementary Fig. 1a). This screen revealed only one As(III) tolerant mutant (Supplementary Fig. 1b) that we named arsenite tolerant 1 (astol1).

The root growth of astol1 was indistinguishable from the wild type (WT) in the absence of As(III). However, upon As(III) treatment astol1 grew 2-times longer roots than WT (Fig. 1a, b). When grown hydroponically in a growth chamber (light intensity ~300 μmol m$^{-2}$ s$^{-1}$), astol1 segregated into two phenotypes, one with decreased growth compared with WT and the other dying of leaves from tips inwards leading to eventual plant death (Fig. 1c, d, and Supplementary Fig. 1c–f). The two phenotypes were later confirmed to represent the heterozygous (astol1(+/−)) and homozygous (astol1(+/+)) mutant, respectively, based on genotyping of the causal gene. Thus, we used astol1(+/−) in most of the experiments. The root elongation assay also revealed that astol1(+/+) was slightly more tolerant to As(III) than astol1(+/−) (Fig. 1h). Moreover, the roots of astol1(+/−) and astol1(+/+) were more tolerant to arsenate (As(V)), compared to WT (Supplementary Fig. 2a, b). Upon As(III) or As(V) treatment, the shoot As concentration of astol1 (+/−) was not significantly different from WT (Supplementary Fig. 2c, d). However, astol1(+/−) accumulated significantly more As in the roots than WT (Supplementary Fig. 2c, d), implying that the increased As tolerance in astol1 is not caused by a decreased As uptake.

We then investigated the relationship between the phenotypes of growth inhibition observed in astol1 and the light intensity in a growth chamber. Increasing light intensity from ~300 to ~800 μmol m$^{-2}$ s$^{-1}$ increased the growth of both WT and astol1(+/−), and decreased the difference in plant biomass between the two genotypes from 63% to 31%, but did not improve the growth of astol1(+/+) (Supplementary Fig. 3a, b). When grown in an open paddy field, where light intensity often exceeds 1000 μmol m$^{-2}$ s$^{-1}$, astol1(+/−) was indistinguishable from WT in all agronomic traits including grain yield (Fig. 1e, Supplementary Fig. 4a–f). Remarkably, in open paddy fields astol1(+/+) also grew to maturity but produced smaller grain yield due to a low percentage of filled spikelets. These results indicate that the slow growth phenotype of astol1 can be reversed under normal light conditions in paddy fields.

### OsASTOL1 encodes a chloroplast-localized OAS-TL protein. To clone the causal gene for the astol1 phenotype, we backcrossed astol1(+/−) (paternal) with WT (maternal). Among the 30 F$_1$ plants, 40% showed the As(III) tolerance phenotype (not significantly different from the expected 50%), indicating that the mutation in OsASTOL1 has a dominant inheritance for As(III) tolerance. In the F$_2$ progenies a segregation pattern with a ratio of 58 WT: 166 astol1(+/−) :15 astol1(+/+) was obtained, suggesting a semi-dominant effect of the mutation. However, the determined ratio was inconsistent with the expected segregation ratio of 1: 2: 1 ($\chi^2 = 51.66 > \chi^2_{0.05,2}$), likely due to partial sterility and poor seed germination of the homozygous mutant.

We conducted genomic resequencing mapping and MutMap analysis[32] on the backcrossed F$_2$ progeny with astol1(+/+) phenotype. One SNP (SNP-26700810) was identified with a SNP index of 1 (Supplementary Fig. 5). This SNP was located in the fifth exon of the gene (Os12g0625000) encoding a putative OAS-TL that leads to a Ser189Asn mutation (codon AGC → AAC, Fig. 1f). The point mutation in this OsASTOL1 gene was verified by DNA sequencing and a derived cleaved amplified polymorphic sequence (dCAPS) marker (Supplementary Fig. 6a, c, d).

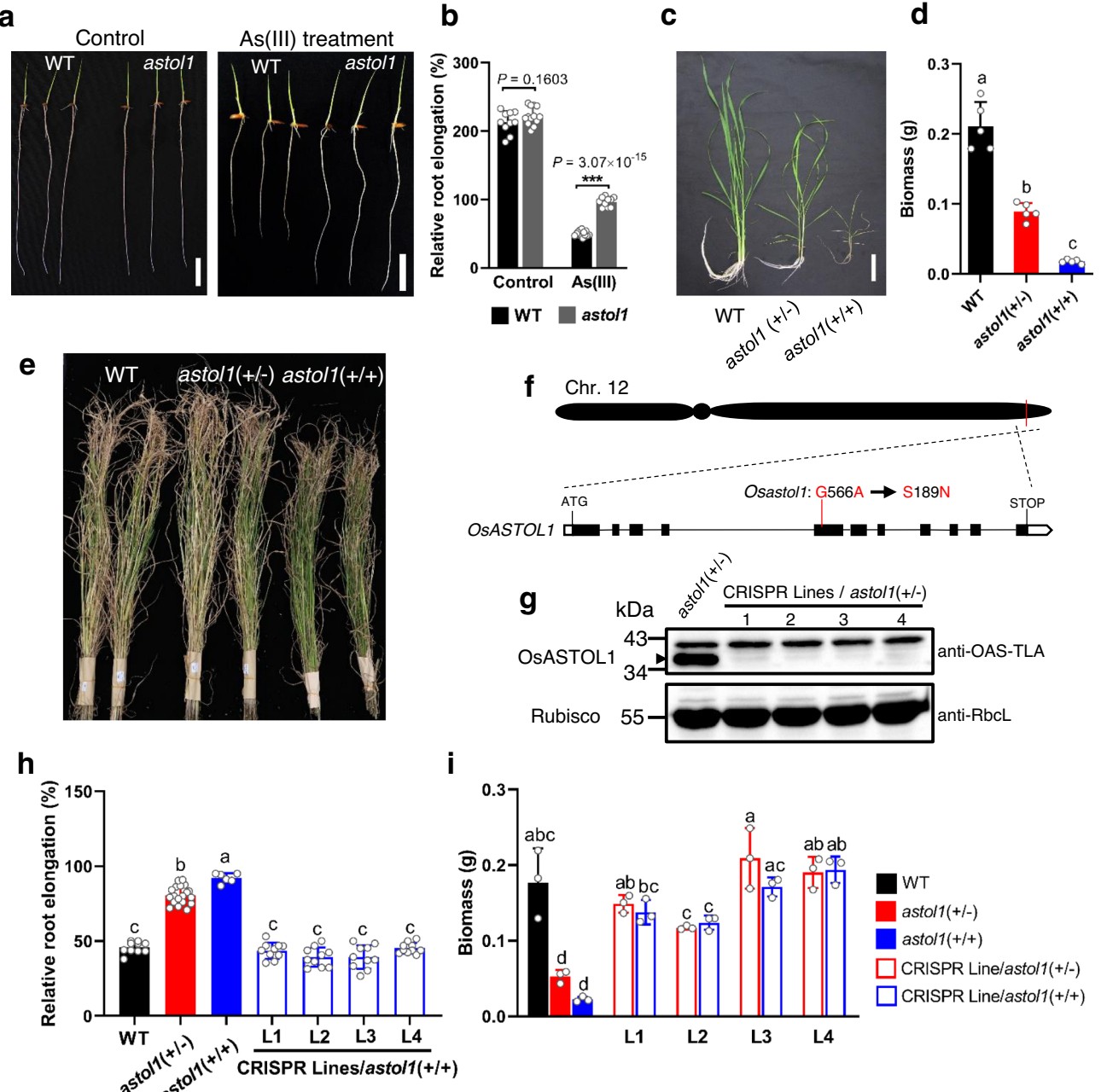

**Fig. 1 Phenotypes of *astol1* mutant and MutMap-based cloning of *OsASTOL1*. a** The phenotype of WT and *astol1* mutant treated with 0 or 20 μM As(III) for 2 days. Three plants for each genotype are shown. Scale bars, 20 mm. **b** Relative root elongation of WT and *astol1* mutant treated with 0 or 20 μM As (III) for 2 days. Asterisks indicate significant difference by two-sided Student's *t* test: ****P* < 0.001. **c** Growth phenotype of 5-week-old WT, *astol1*(+/−) and *astol1*(+/+) mutant grown hydroponically. Two plants for each genotype are shown. Scale bars, 10 cm. **d** Biomass of 5-week-old WT, *astol1*(+/−) and *astol1*(+/+) mutant grown hydroponically. **e** Wild-type rice and *astol1* mutants were grown in a paddy field until maturity. **f** Chromosomal location and gene structure of *OsASTOL1*. White and black boxes represent untranslated regions and exons, respectively, and the lines between them represent introns. Red line indicates the mutation site. **g** Validation of knock-out OsASTOL1/ OsASTOL1$^{S189N}$ protein in four independent CRISPR lines in the *astol1*(+/−) background using immunoblot analysis. OsASTOL1 protein (lower band) is detected by AtOAS-TL A1 antibody (anti-OAS-TLA) and the Rubisco large subunit is used as a loading control. The experiment was repeated three times and similar results were obtained. **h** Relative root elongation of WT, *astol1* (+/−), *astol1*(+/+), and four independent CRISPR lines in the *astol1*(+/+) background. Plants were treated with 20 μM As(III) for 2 days. **i** Biomass of 5-week-old WT, *astol1*(+/−), *astol1*(+/+), and CRISPR lines in *astol1*(+/−) or *astol1*(+/+) background. DW, dry weight. WT, wild type. *astol1*(+/−), *astol1* heterozygote. *astol1*(+/+), *astol1* homozygote. Data in **b**, **d**, **h**, and **i** are shown as means ± s.d., *n* = 10 for WT (control) and *astol1* (As(III) treatment), 13 for *astol1* (control) and WT (As(III) treatment) (**b**), 5 (**d**), 6 for *astol1*(+/+), 10 for WT and all CRISPR lines/*astol1*(+/+), 19 for *astol1*(+/−) (**h**) or 3 (**i**) biological replicates; each biological replicate represents an individual plant. Different letters in **d**, **h**, and **i** indicate significant differences (*P* < 0.05) using one-way ANOVA followed by Tukey's test.

Since *astol1* is a semi-dominant mutant, we generated transgenic plants expressing the mutant *Osastol1* gene under the control of the maize *Ubiquitin* promoter in WT. When grown in an open paddy field, all transformed $T_0$ individual lines ($n = 5$) that over-expressed the mutant OsASTOL1$^{S189N}$ protein produced sterile seeds (Supplementary Fig. 7). This finding supports our previous observation that *astol1*(+/+) showed a low percentage of filled spikelets in the field and induces plant death when grown in growth chambers (Fig. 1d, Supplementary Fig. 4d). To provide direct evidence that mutated *OsASTOL1* is responsible for the *astol1* phenotypes, we knocked out the mutant *Osastol1* gene in *astol1* using the CRISPR-Cas9 gene-editing method (Supplementary Fig. 6b). Four independent $T_0$ CRISPR-induced knockout lines in the *astol1*(+/−) background with both *OsASTOL1* and *Osastol1* being homozygously knocked out were obtained and confirmed by immunoblot using an antibody against the full-length *Arabidopsis* OAS-TL A (Lower band in Fig. 1g, with the upper band likely to be another isoform of OAS-TL). In the $T_1$ generation of these lines, three types of plants including homozygous knockout of *OsASTOL1* and/or *Osastol1* in the backgrounds of WT, *astol1*(+/−) or *astol1*(+/+) were isolated by using the dCAPS marker (Supplementary Fig. 6d). The relative root elongation of all CRISPR lines in the *astol1*(+/+) background showed the same As(III) sensitive phenotype as WT (Fig. 1h). Furthermore, all CRISPR lines in both the *astol1*(+/−) and *astol1*(+/+) backgrounds showed a complete rescue of the *astol1* mutation-induced phenotypes in growth and root As concentration (Fig. 1i and Supplementary Fig. 6c–e). These results demonstrate that the S189N point mutation in the OsASTOL1 protein is the cause for the *astol1* phenotypes.

OsASTOL1 shares high homology with *Arabidopsis* cytosolic OAS-TL A (Supplementary Fig. 8). However, in contrast to AtOAS-TL A, OsASTOL1 contains 87 extra amino acid residues at the N-terminus (Supplementary Fig. 9), which are predicted to generate a chloroplastidic transitpeptide[33] (Supplementary Table 2). *OsASTOL1* is predominately expressed in photosynthetic tissues (Supplementary Fig. 10a–d). When transiently expressed in tobacco leaves and rice protoplasts, both wild-type OsASTOL1-YFP and mutant OsASTOL1$^{S189N}$-YFP fusion proteins localize in the chloroplasts (Fig. 2a and Supplementary Fig. 10e). These findings demonstrate that the *astol1* point mutation does not affect the chloroplastic localization of OsASTOL1 in rice.

**Ser189 of OsASTOL1 is crucial for OAS-TL activity**. To test whether the OsASTOL1$^{S189N}$ mutation affects OAS-TL activity *in planta*, we determined the total OAS-TL activity and OAS-TL protein abundance in the WT and the *astol1* mutant. Remarkably, the abundance of the mutated OsASTOL1$^{S189N}$ was unaffected in *astol1* (+/+), but the total extractable OAS-TL activity was decreased by 32% in *astol1*(+/+) when compared to WT (Fig. 2b, c). These results suggest that Ser189Asn mutation does not destabilize the OsASTOL1$^{S189N}$ protein but impairs its enzymatic activity.

OAS-TL proteins are highly conserved in prokaryotes and eukaryotes (Supplementary Figs. 9 and 11). In particular, the Ser189 of OsASTOL1 is conserved in all analyzed OAS-TLs (Supplementary Fig. 9). In order to compare the OAS-TL activity of the mature OsASTOL1 and the mutated OsASTOL1$^{S189N}$, we deleted the 87 amino acid residues encoding the transitpeptide and expressed them in *E. coli*. Because AtOAS-TL A has been well characterized in enzymatic analysis[23,26] and is highly homologous to OsASTOL1 protein (Supplementary Fig. 8), it was applied as a positive control for a canonical OAS-TL enzyme (Fig. 2d). Expression of AtOAS-TL A in *E. coli* Cys- auxotrophic strain

NK3[34], which lacks the cysteine synthase gene and consequently is unable to grow on medium without cysteine supplement, restored its growth in Cys-free medium within 2 days of incubation (Supplementary Fig. 12a). In comparison, expression of OsASTOL1 restored the growth of NK3 strain in Cys-free medium after 7 days of incubation, whereas OsAS-TOL1$^{S189N}$ failed to complement NK3 strain (Supplementary Fig. 12a). Moreover, the purified mature OsASTOL1 protein clearly displayed OAS-TL activity, but its specific activity was only about 1/4 of AtOAS-TL A (Fig. 2e). In contrast, the mutated OsASTOL1$^{S189N}$ displayed no OAS-TL activity (Fig. 2e). This is consistent with the complementation assay in NK3 (Supplementary Fig. 12a). Moreover, size exclusion chromatography (SEC) analysis shows that both the OsASTOL1 and OsASTOL1$^{S189N}$ were present as dimers as other OAS-TLs (Supplementary Fig. 12b). Furthermore, the replacement of the corresponding Ser102 in AtOAS-TL A with Asn fully inactivated the OAS-TL A$^{S102N}$ mutant protein (Fig. 2e). The protein crystal structure[35] of AtOAS-TL A indicates that Ser102 is located at the opening of the binding site for the substrate OAS (Supplementary Fig. 13a, b).

To investigate the role of this conserved Ser, we changed the Ser189 of OsASTOL1 to different amino acid residues by site-directed mutagenesis and determined the in vitro OAS-TL activities of the respective protein mutants (Supplementary Fig. 14a, b). All mutation alleles lost their enzymatic activity, except OsASTOL1$^{S189T}$ that displayed 41% of the ASTOL1 activity (Supplementary Fig. 14b). Because both the mutation alleles of S189A (a constitutively nonphosphorylable form) and S189D (a mimic of the constitutively phosphorylated form) lost the OAS-TL activity completely, it is unlikely that phosphorylation of S189 is required for the enzyme activity. Thus, we conclude that the conserved Ser189 in OsASTOL1 is crucial for OAS-TL activity, and mutation of this residue in OsASTOL1$^{S189N}$ leads to a total loss of OAS-TL activity.

**Loss of OAS-TL activity is irrelevant to *astol1* phenotype**. To investigate the causal relationship between the phenotypes of the *astol1* mutation and the OAS-TL activity in the mutant, we generated CRISPR lines and overexpression lines of *OsASTOL1* in the WT background. The CRISPR-induced loss-of-ASTOL1 function resulted in three independent lines in a decrease to 50–63% of the total in vivo OAS-TL activity when compared to WT (Supplementary Fig. 15a, b). In contrast, overexpression of wild type ASTOL1 protein caused in three independent lines a 3-fold increase of OAS-TL activity when compared to WT (Supplementary Fig. 15b). Neither loss-of-ASTOL1 function nor the overexpression of ASTOL1 mimicked the As(III) tolerance or the decreased growth phenotype of *astol1* (Supplementary Fig. 15c, d). These results suggest that the *astol1* phenotypes are not caused by the loss of OAS-TL activity. This finding is consistent with the semi-dominant inheritance of the *astol1* mutation. The results are also consistent with previous studies on *Arabidopsis* that knock-out of each of the three major isoforms of OAS-TL did not impair growth[25].

**Stabilized CSC leads to higher SAT activity in *astol1* mutant**. In plants, Cys biosynthesis is not limited by OAS-TL, but by the synthesis of its substrate OAS[25,26,36]. OAS is exclusively produced by SAT in plants. Remarkably, SATs are almost not regulated at the transcriptional level. The dominant mechanism for the regulation of SAT activity is the formation of the CSC complex. CSC formation significantly activates SAT and is controlled by the actual OAS supply[22]. We applied the well-characterized AtSAT5 protein as the bait protein to test whether Ser189Asn mutation in the mutant OsASTOL1$^{S189N}$ protein affects its association with

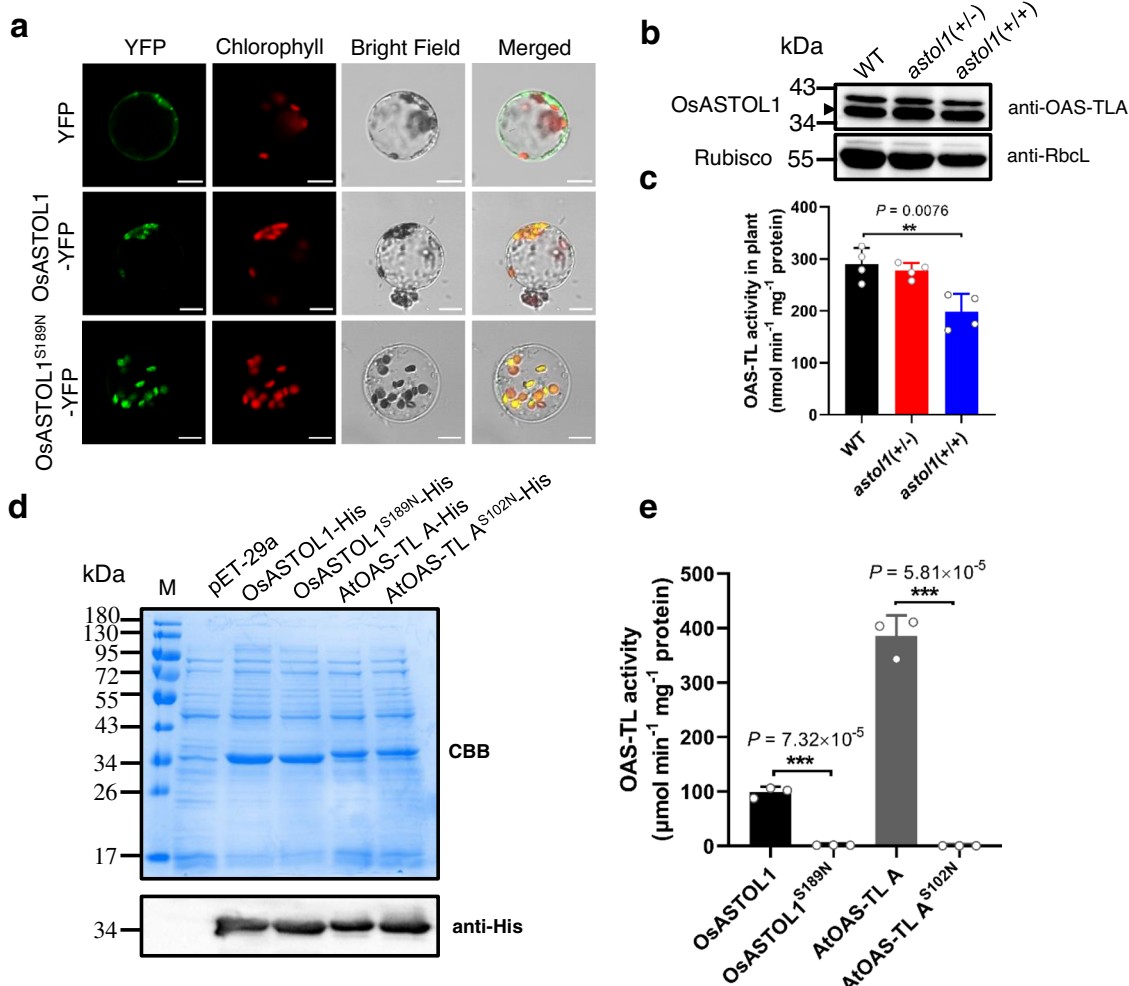

**Fig. 2 *OsASTOL1* encodes a chloroplast-localized *O*-acetylserine(thiol)lyase (OAS-TL). a** Subcellular localization of OsASTOL1. Rice protoplasts were expressed with the *eYFP* (top), *OsASTOL1-eYFP* (middle), or *Osastol1-eYFP* (bottom) driven by the cauliflower mosaic virus 35S promoter. Left to right: YFP fluorescence, chlorophyll autofluorescence, bright-field images, and merged images. Scale bars, 10 μm. At least five protoplasts were investigated with similar results to the images shown in **a**. **b** Validation of OsASTOL1/ OsASTOL1$^{S189N}$ protein (lower band) levels in 3-week-old WT, *astol1*(+/−), and *astol1* (+/+) using immunoblot analysis. OsASTOL1/ OsASTOL1$^{S189N}$ protein levels are detected by AtOAS-TL A1 antibody (anti-OAS-TL A) and Rubisco large subunit is used as a loading control. The experiment was repeated two times and similar results were obtained. **c** Total OAS-TL enzyme activity in whole soluble protein extracts of the shoots of 3-week-old WT, *astol1*(+/−), and *astol1*(+/+). Data are shown as means ± s.d., *n* = 4 biological replicates; each biological replicate represents an individual plant. **d** Expression of recombinant mature OAS-TL proteins in *E. coli*. Proteins were separated on SDS-PAGE and visualized by Coomassie Brilliant Blue (CBB) staining (upper) and by immunoblot with anti-His antibody (bottom). M, marker. The experiment was repeated two times and similar results were obtained. **e** In vitro OAS-TL enzyme activity of purified mature OsASTOL1 and AtOAS-TL A and their corresponding mutant proteins. Data are shown as means ± s.d., *n* = 3 technical replicates. WT, wild type. *astol1*(+/−), *astol1* heterozygote. *astol1*(+/+), *astol1* homozygote. Asterisks in **c**, **e** indicate significant difference by two-sided Student's *t* test: **$P < 0.01$, ***$P < 0.001$.

SAT protein in an affinity column-based pull-down experiment[29,37]. Both OsASTOL1 and OsASTOL1$^{S189N}$ proteins interact with the His-tagged AtSAT5 protein (Supplementary Fig. 16a, b). Similarly, AtOAS-TL A and AtOAS-TL A$^{S102N}$ bound to the column immobilized His-tagged AtSAT5 (Supplementary Fig. 16c, d). Thus, mutation of the conserved Ser to Asn did not affect the formation of CSC.

Next, we tested the ability of OAS to dissociate the recombinant CSCs consisting of His-SAT5 and the four candidate OAS-TLs. As expected, the CSC containing the wild type OsASTOL1 and the wild type AtOAS-TL A were dissociated by the application of OAS. In contrast, OAS failed to dissociate the CSC consisting of the OsASTOL1$^{S189N}$ and the AtOAS-TL A$^{S102N}$ protein (Fig. 3a–c, and Supplementary Fig. 16e–g). Since OAS-induced dissociation of the CSC is triggered by competition between OAS and the SAT C-terminus for binding to the active

site of OAS-TL[38,39], we quantified the affinity of wild type OsASTOL1 and the OsASTOL1$^{S189N}$ for OAS and the C-terminal fragment of SAT5 (AtSAT5C10) by microscale thermophoresis (MST). As expected, wild type ASTOL1 displayed high affinity towards OAS and AtSAT5C10 (Fig. 3d, e and Supplementary Table 3). The mutated OsASTOL1$^{S189N}$ physically interacted with the AtSAT5C10 fragment with a wild-type like affinity, but specifically lost the affinity towards OAS (Fig. 3d, e and Supplementary Table 3). Mutation of S102N in AtOAS-TL A resulted in the same specific loss of affinity towards OAS but did not affect the binding of AtSAT5C10 (Supplementary Fig. 16h, i and Supplementary Table 3). These biochemical properties of the mutated OAS-TLs provide a molecular explanation for the formation of an OAS-resistant CSC by the mutated OsAS-TOL1$^{S189N}$ and AtOAS-TL A$^{S102N}$ proteins and the catalytic inactivity of the both proteins.

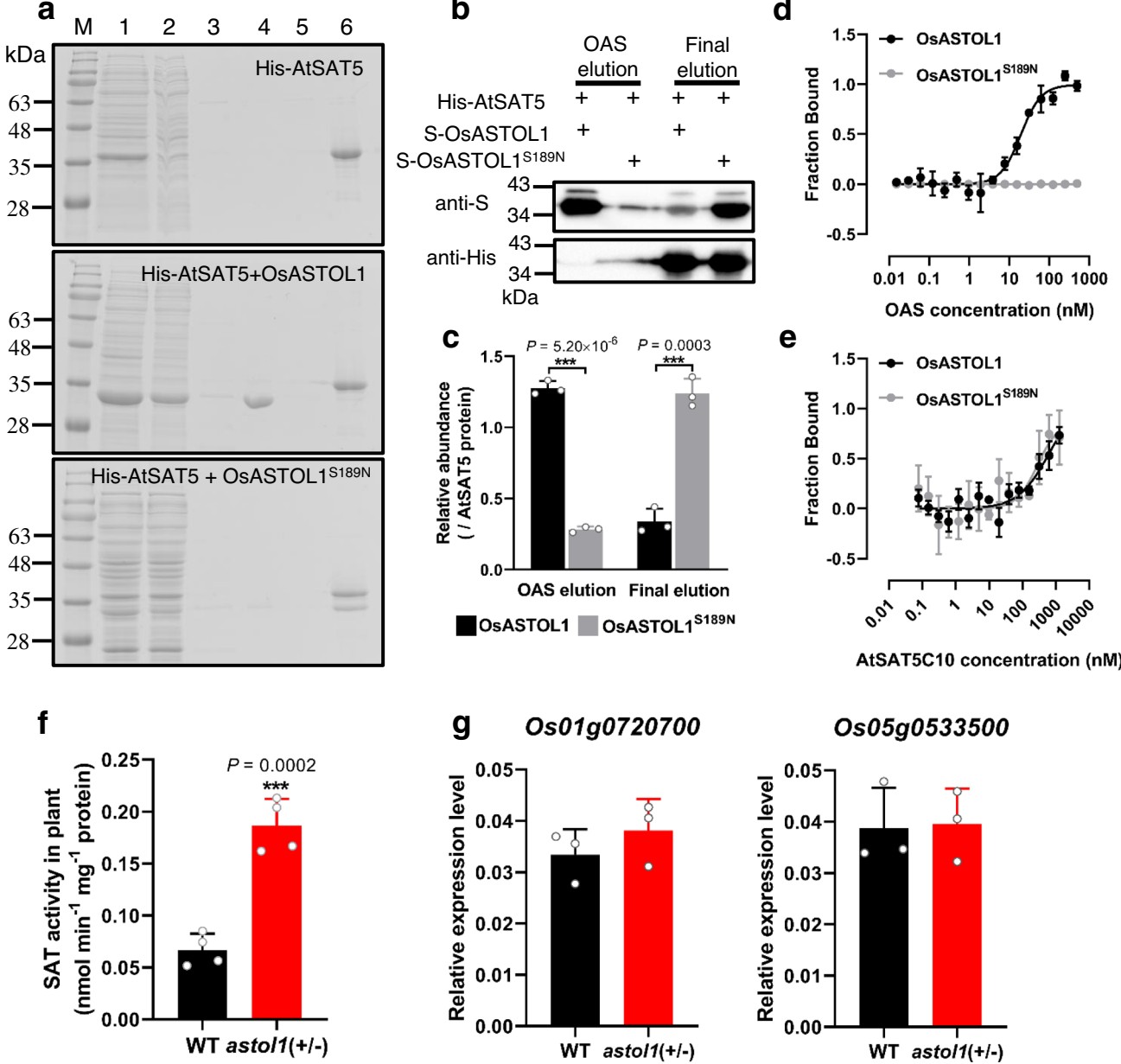

**Fig. 3 astol1 mutant increases SAT activity by stabilizing cysteine synthase complex (CSC). a** In vitro pull-down analysis of the dissociation effect of OAS on CSC. The fraction of different elution steps (lanes 1–6) from His-AtSAT5 protein (upper panel), His-AtSAT5 and mature OsASTOL1 protein (middle panel), and His-AtSAT5 and mature OsASTOL1$^{S189N}$ protein (bottom panel) was analyzed by SDS-PAGE and visualized by Coomassie Brilliant Blue staining. M marker; lane 1: crude extract of His-AtSAT5 (upper), OsASTOL1 (middle) and OsASTOL1$^{S189N}$ (bottom); lane 2: flow through of His-AtSAT5 (upper), OsASTOL1 (middle) and OsASTOL1$^{S189N}$ (bottom); lane 3: washing buffer (80 mM imidazol); lane 4: OAS elution (10 mM OAS + 80 mM imidazol); lane 5: washing buffer (80 mM imidazol); lane 6: final elution (400 mM imidazol). The experiment was repeated three times and similar results were obtained. **b** The fraction of OAS elution and final elution was analyzed by SDS-PAGE and visualized by immunoblot, and a representative result of three repeated tests is shown. **c** The relative abundance of OsASTOL1 or OsASTOL1$^{S189N}$ in the OAS elution and the final elution was quantified by ImageJ software, compared to the corresponding AtSAT5 protein. Microscale thermophoresis (MST) analysis of in vitro binding of OsASTOL1 and OsASTOL1$^{S189N}$ protein with OAS (**d**) or AtSAT5C10 peptide (**e**). **f** Total SAT enzyme activity in whole soluble protein extracts of the shoots of 3-week-old WT and astol1 (+/−). **g** Relative expression levels of two rice SAT genes (Os01g0720700 and Os05g0533500) in the shoots of 3-week-old WT and astol1(+/−), with OsHistone H3 as the internal reference. WT, wild type. astol1(+/−), astol1 heterozygote. Data in **c–e** are shown as mean ± s.d., $n = 3$ technical replicates. Data in **f–g** are shown as means ± s.d., $n = 4$ (**f**) or 3 (**g**) biological replicates; each biological replicate represents an individual plant. Asterisks in **c**, **f**, **g** indicate significant difference by two-sided Student's t test: **$P < 0.01$, ***$P < 0.001$.

To test whether the endogenous SAT activity was stimulated by the presence of mutated OsASTOL$^{S189N}$ protein in astol1, we compared SAT activities in crude plant protein extracts of WT and the astol1(+/−) mutant. The extractable SAT activity of astol1 (+/−) was 2.8-fold higher than that in WT (Fig. 3f), although

transcript abundance of SAT genes was indistinguishable between astol1(+/−) and WT (Fig. 3g). Taken together, these results imply that the Ser189Asn mutation in OsASTOL1$^{S189N}$ increases the stability of the CSC against the dissociation of OAS, resulting in significantly enhanced endogenous SAT activity in planta.

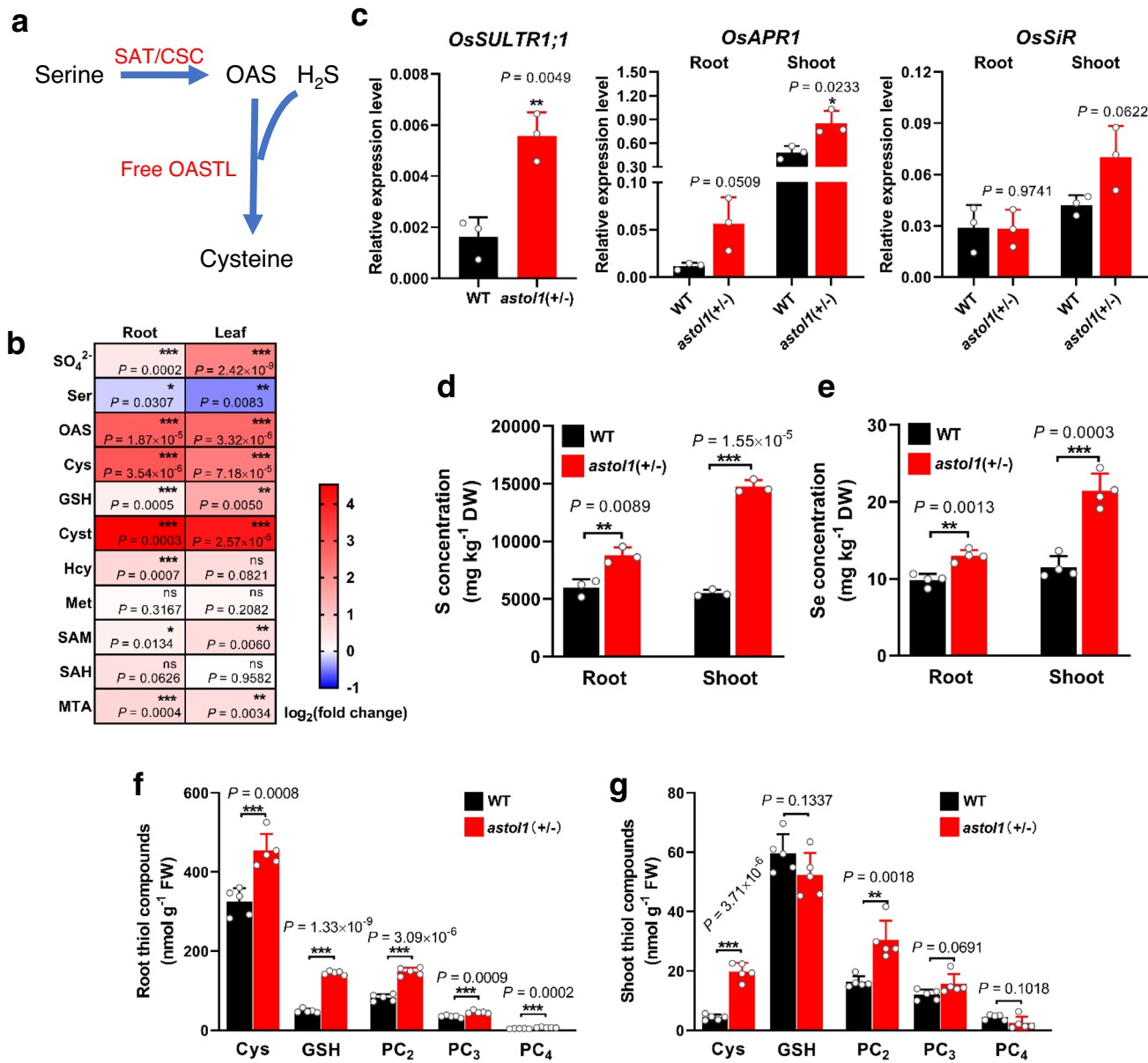

**Fig. 4 astol1 mutant enhances S and Se uptake and phytochelatins-dependent As detoxification. a** Metabolic pathway from serine to cysteine. **b** Relative fold change of S-related metabolites in astol1(+/−) compared to WT, visualized as the heat map (n = 5 biological replicates for sulfate, three biological replicates for all other metabolites; each biological replicate represents an individual plant). Abbreviations of metabolites: $SO_4^{2-}$ sulfate, Ser serine, OAS O-acetylserine, Cys cysteine, Cyst cystathionine, GSH glutathione, Hcy homocysteine, Met methionine, SAH S-adenosylhomocysteine, SAM S-adenosylmethionine, MTA methylthioadenosine. **c** Relative expression levels of rice genes involved in sulfate uptake and reduction in the roots and shoots of 3-week-old WT and astol1(+/−). Transcriptional induction of selected rice genes (OsSULTR1;1 (Os03g0195800) in roots, OsAPR1 (Os07g0509800) and OsSiR (Os05g0503300) in roots and shoots) was measured by Q-PCR, with OsHistone H3 as the internal reference gene. **d** S concentrations in roots and shoots of 5-week-old WT and astol1(+/−). **e** Se concentrations in roots and shoots of 5-week-old WT and astol1(+/−) exposed to 2 μM Se(VI) for 3 days. The concentrations of non-protein thiols in roots (**f**) and shoots (**g**) of 4-week-old WT and astol1(+/−) treated with 5 μM As(III) for 3 days. DW, dry weight. FW, fresh weight. WT, wild type. astol1(+/−), astol1 heterozygote. Data in **c–g** are shown as means ± s.d. n = 3 (**c**, **d**), 4 (**e**) or 5 (**f**, **g**) biological replicates; each biological replicate represents an individual plant. Asterisks in **b–g** indicate significant differences by two-sided Student's t test: *P < 0.05, **P < 0.01, ***P < 0.001.

**astol1 mutant overaccumulates OAS and S metabolites**. As a result of the increased foliar SAT activity, the total steady-state level of the SAT product, OAS, Cys and most of the Cys-derived sulfur-metabolites tested were significantly higher in leaves of astol1(+/−) when compared to WT. Remarkably, the astol1 mutation-induced increase of OAS and sulfur-containing metabolites were also detected in roots of non-As(III)-stressed plants, suggesting that SAT activity is globally enhanced by the astol1 mutation (Fig. 4a, b and Supplementary Table 4). The astol1

mutation also caused decreased steady-state level of the SAT substrate, serine, in leaves and roots, suggesting a significant higher conversion of Ser into OAS in these organs (Fig. 4a, b and Supplementary Table 4).

OAS acts as the signal of sulfur deficiency and induces the expression of genes involved in S uptake and assimilation[21,40,41]. The transcript levels of several genes involved in sulfate uptake and reduction, including root plasma membrane-localized high-affinity sulfate transporter OsSULTR1;1, were substantially

upregulated in *astol1* when compared with WT (Fig. 4c). Consistently, the *astol1*(+/−) mutant accumulated more sulfate, total S and Se (Fig. 4b, d, e); the latter is a chemical analog of S and is also taken up by SULTRs. Moreover, knockout of *Osastol1* in the *astol1* background reduced the shoot S concentration to the WT level, whereas knockout of wild-type *OsASTOL1* did not significantly affect shoot S concentration (Supplementary Figs. 6f and 15e), further supporting that the specific mutation in *Osastol1* leads to the mutant phenotypes in *astol1*. Taken together, these results demonstrate that substantial accumulation of OAS leads to increased sulfate uptake and assimilation into Cys in the *astol1* mutant.

To explain the OsASTOL1^S189N-induced gain of As(III) tolerance, we quantified the impact of As(III) on thiol compounds in the roots and shoots of WT and *astol1*(+/−). Upon 5 μM As (III) treatment, the steady-state levels of Cys, and the As(III) chelators GSH, $PC_2$, $PC_3$, and $PC_4$ were all significantly increased in the roots of *astol1*(+/−) when compared with WT (Fig. 4f), demonstrating the enhanced capacity of the *astol1* plants for complexation of As(III). The levels of Cys and $PC_2$ in the shoots of *astol1*(+/−) were also increased significantly (Fig. 4g), although the levels of both thiols were lower than those in the roots. Furthermore, the transcript levels of *OAS-TL* and *SAT* genes in shoots and roots of WT and *astol1*(+/−) showed no significant change when treated with As(III) (Supplementary Fig. 17a–d), suggesting that these genes are not transcriptionally induced by As(III) stress.

**OsASTOL1^S189N reduces As and increases Se in rice grain.** To evaluate the biotechnological capability of the *astol1* mutation for improvement of the nutritional value of rice grain, we quantified the yield and accumulation of As, S, and Se in the grain of rice plants grown on paddy fields with background levels of As and Se in the soils. Grain As concentration of *astol1*(+/−) was 29% and 33% lower than that of WT, respectively, in the two field experiments (Fig. 5a and Supplementary Fig. 18a). In contrast, *astol1*(+/−) accumulated 77% and 107% higher Se concentration in rice grain than WT, respectively, in the two field experiments (Fig. 5c and Supplementary Fig. 18b). We also determined As concentrations in different shoot tissues of *astol1*(+/−) and WT sampled at plant maturity from the two field sites. The *astol1*(+/−) mutant contained significantly higher As concentrations in node I and internode than WT at one site, but a significantly lower As concentration in flag leaf blade at the other site (Supplementary Fig. 19a, b).

To overcome the growth inhibition in *astol1*(+/+), we overexpressed wild-type *OsASTOL1* in *astol1*(+/−) mutant and obtained overexpression lines in the WT, *astol1*(+/−), or *astol1*(+/+) backgrounds among the segregated T$_1$ plants. The overexpression lines in the *astol1*(+/+) background rescued the early leaf death phenotype of *astol1*(+/+) and also improved the growth of *astol1*(+/−) under low light conditions (Supplementary Fig. 20a–f). Notably, the overexpression lines in both the backgrounds of *astol1*(+/−) and *astol1*(+/+) still accumulated higher S concentrations in the shoots and higher As concentrations in the roots than WT (Supplementary Fig. 20g–l). Moreover, when grown in a paddy field, overexpression of wild-type *OsASTOL1* in both the *astol1*(+/−) and *astol1*(+/+) backgrounds restored their growth and agronomic traits to the levels of WT (Fig. 5d, e and Supplementary Fig. 21a–e). Furthermore, the overexpression lines in both backgrounds of *astol1*(+/−) and *astol1*(+/+), like *astol1*(+/−), had significantly lower grain As concentrations than WT, with decreases of 32–39% and 23–40%, respectively (Fig. 5a). These lines accumulated significantly more S in the grain than WT (Fig. 5b). In addition, the overexpression lines in the backgrounds of *astol1*(+/−) and *astol1*(+/+) accumulated 95–121% and 15–48% higher Se concentration in the grain than WT, respectively (Fig. 5c). These results indicate that the Ser189Asn mutation in OsASTOL1^S189N in rice enriches S and Se and reduces As accumulation in rice grain, while the growth inhibition in *astol1*(+/+) can be overcome by balancing the mutant and wild-type OsASTOL1 protein.

## Discussion

Rice accumulates about 10-times higher As in the grain than other cereal crops due to the mobilization of As(III) in submerged paddy soils and the high capacity of rice roots for uptake of As (III)[3,42,43]. Consequently, many Asian populations are exposed to highly levels of dietary As. On the other hand, around 75% of the rice grains produced globally have insufficient levels of Se to meet human's requirement[44]. In this study, we show that Ser189Asn mutation in OsASTOL1^S189N serves as a molecular switch in the sulfur assimilation pathway to produce multiple desirable consequences, including increased As tolerance, decreased As accumulation in the grain and increased Se accumulation in the grain. These phenotypes can be explained by improved uptake and assimilation of sulfate and selenate, and the enhanced synthesis of thiol compounds. This reinforced capacity to synthesize PCs allows for improved complexation of As[11–14], facilitates the vacuolar sequestration of As-thiol complexes and, consequently, reduces As mobility to rice grain[15–17]. Increased synthesis of thiol compounds is more pronounced in the roots than in the shoots of the mutant, which is consistent with roots being the main tissue for As sequestration. A radioactive $^{73}$As tracer experiment showed that approximately 90% of the arsenite taken up by rice roots is sequestered in the roots[45]. Furthermore, enhanced GSH and PC synthesis should also improve the resistance towards heavy metals like Cd and Hg as well as some biotic stresses[18,46,47].

We show that Ser189Asn mutation in OsASTOL1^S189N results in a total loss of the OAS-TL activity but a gain-of-function of increased SAT activity, and it is the latter that explains the observed phenotypes of the *astol1* mutant, as summarized in our model (Fig. 6). The Ser189Asn mutation in OsASTOL1^S189N specifically impairs its affinity towards the substrate OAS (Fig. 3d), but surprisingly does not impact its physical interaction with the SAT5C10 peptide (Fig. 3e). In fact, the CSC complex formed from OsASTOL1^S189N and SAT becomes much more resistant to the dissociation of OAS (Fig. 3a–c), which may also result from the loss of binding of OAS to OsASTOL1^S189N. The increased stability of the CSC complex leads to a significantly higher SAT activity in the *astol1* mutants, because SAT is substantially activated in the CSC complex due to release from cysteine inhibition[48,49]. Because production of OAS is the rate-limiting step, increased OAS drives Cys biosynthesis catalyzed by other isoforms of OAS-TL either in the chloroplast or in the cytoplasm, as proposed in the model (Fig. 6). Moreover, OAS is a signaling molecule that can induce the expression of genes involved in S uptake and assimilation[21,40,41]. Indeed, several genes in the sulfate uptake and reduction pathway were significantly upregulated in the *astol1* mutants (Fig. 4c). In addition to sulfate, many sulfur-containing metabolites are also increased in the *astol1*(+/−) mutant, indicating that sulfate reduction and assimilation are enhanced (Fig. 4b). Increased biosynthesis of OAS, Cys, and other elevated thiol compounds in the *astol1* mutants causes a higher demand for carbon. Consequently, norm light conditions in paddy fields helped the mutants to overcome the imposed challenge for the enhanced synthesis of the diverse carbon and sulfur-containing metabolites. The relatively poor growth of the mutants under low light intensity in growth rooms might be also partially explained by a misbalance of enhanced

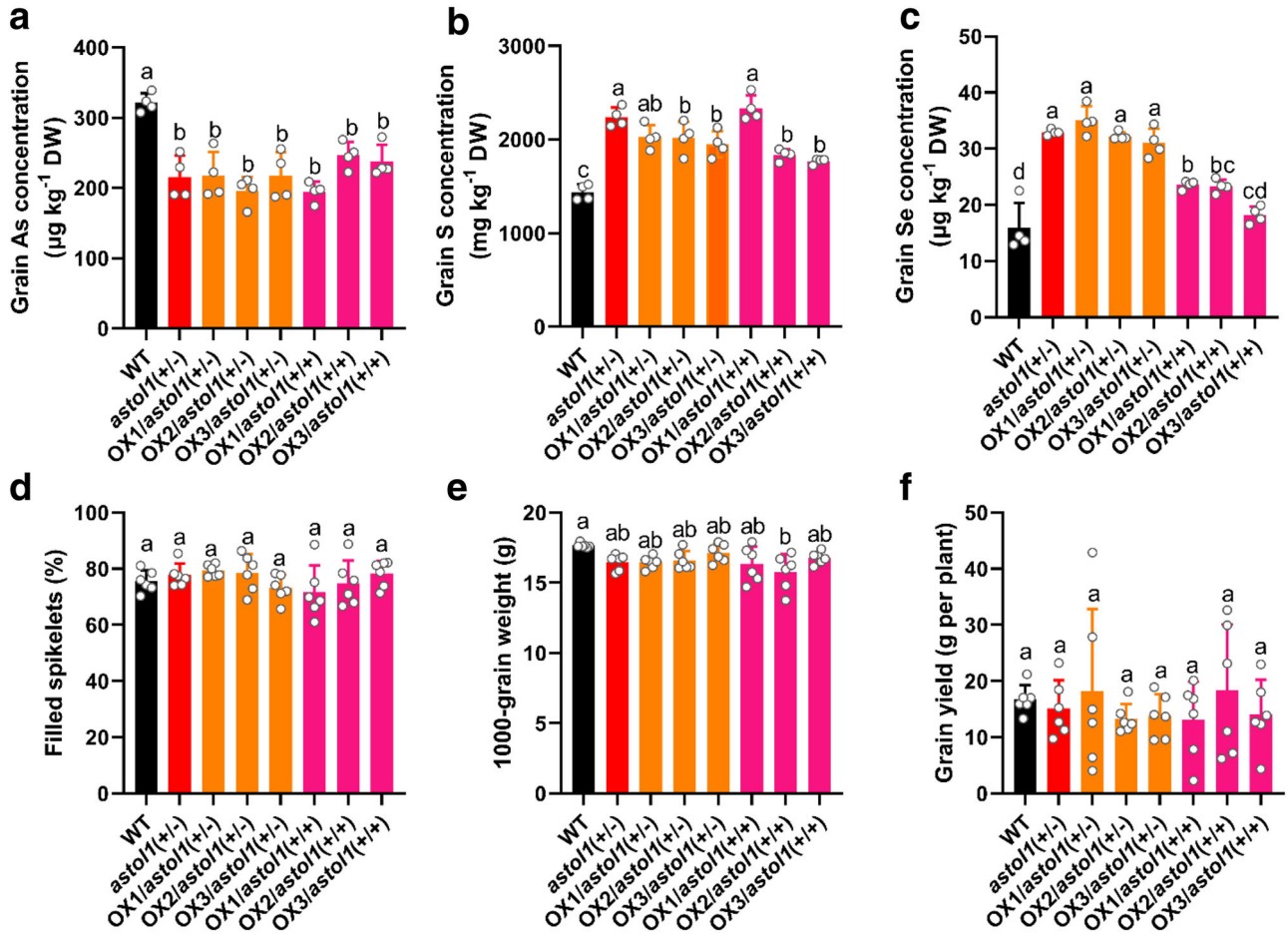

**Fig. 5 Mutation allele of OsASTOL1 reduces As and increases S and Se concentrations in rice grain.** Grain As (**a**), S (**b**), and Se (**c**) concentrations of WT, *astol1*(+/−), and three independent *OsASTOL1* overexpression lines in the *astol1*(+/−) or *astol1*(+/+) background. Plants were grown in a paddy field of Nanjing (12.5 mg kg⁻¹ As and 0.61 mg kg⁻¹ Se in the soil). **d–f** Yield components of WT, *astol1*(+/−), and three overexpression lines in *astol1*(+/−) or *astol1*(+/+) background. After harvest, filled spikelets (**d**), 1000-grain weight (**e**), and grain yield (**f**) were recorded. DW, dry weight. WT, wild type. *astol1* (+/−), *astol1* heterozygote. *astol1*(+/+), *astol1* homozygote. Data in **a–f** are shown as means ± s.d., $n = 4$ (**a–c**) or 6 (**d–f**) biological replicates; each biological replicate represents an individual plant. Different letters in **a–f** indicate significant differences ($P < 0.05$) using one-way ANOVA followed by Tukey's test.

level of sulfur-containing defense compounds in the artificial absence of stresses, e.g., high light and pathogens, which demand sulfur-containing defense compounds. However, in open paddy fields the potentially defense-primed *astol1*(+/−) mutants performed as well as the wild type, since the wild type was also challenged with environmental cues inducing sulfur-defense metabolite synthesis.

Our model (Fig. 6) also explains the dosage effect of the mutant allele, as well as the alleviation effect on the mutant growth phenotypes when wild-type *OsASTOL1* was over-expressed in the *astol1* mutants (Fig. 5 and Supplementary Fig. 20), resulting from the balance in the competition between the mutant and wild-type OsASTOL1 for SAT. Taken together, the specific Ser189Asn mutation in OsASTOL1^S189N enables it to act as a regulatory protein of Cys biosynthesis, even though the mutant protein is inactive as an OAS-TL enzyme. It is remarkable that a single amino acid mutation in OAS-TL can trigger such an overwhelming impact on the uptake and assimilation of the macronutrient sulfur, albeit plastidic OAS-TLs is dispensable[25] (Supplementary Fig. 15c–e). The full conservation of Ser189 in all known plant OAS-TLs suggests that the herein identified mechanism to trigger S and Se accumulation in grains will also be applicable

in other plant species. As a proof of concept, we generated an OsASTOL1^S189N-like mutation in cytosolic AtOAS-TL A (AtOAS-TLA^S102N), which, indeed, caused the identical bio-chemical properties as observed in the plastid-localized OsASTOL1^S189N protein of rice (Fig. 2e and Supplementary Fig. 16e–i).

In the reference plant *Arabidopsis thaliana*, cysteine synthesis takes place in all subcellular compartments that are capable of protein biogenesis, which is the dominant sink for cysteine[21]. Remarkably, these subcellular compartments are also equipped with a CSC, which is supposed to regulate the OAS precursor biogenesis by controlling the SAT activity in these compartments. Since 80% of total SAT activity is localized in mitochondria[50,51], the mitochondrial CSC is supposed to efficiently regulate cellular OAS synthesis in plants[49]. Accordingly, downregulation of mitochondrial SAT3 impairs growth of Arabidopsis[52]. However, expression of SAT in the cytosol of *Arabidopsis* and *Nicotiana tabacum* also resulted in enhanced cysteine formation, demon-strating that enhanced OAS production in these compartments also trigger net cysteine synthesis rate[53,54]. In particular, the plastid-localized CSC is supposed to integrate environmental cues to coordinate cysteine biosynthesis for GSH production to maintain redox homeostasis upon diverse stresses[55–57]. In

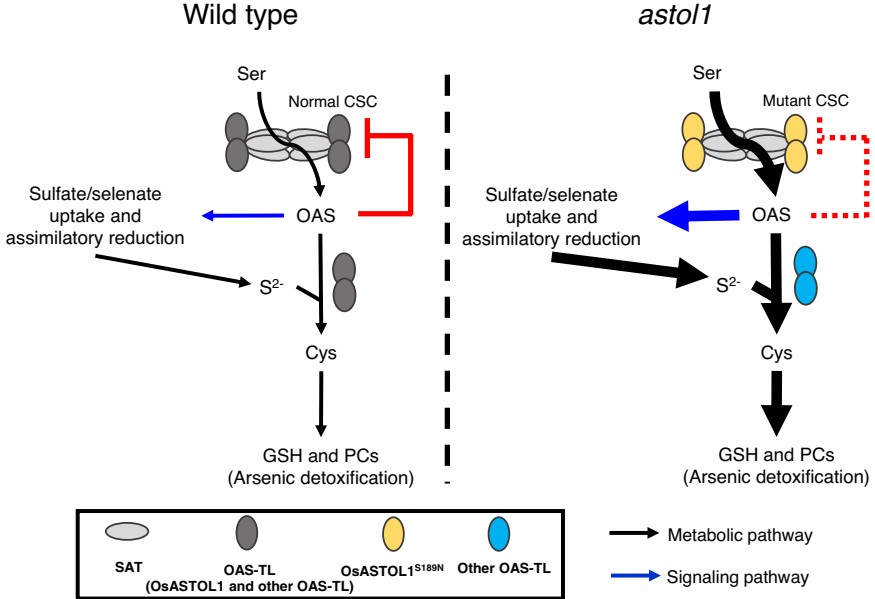

**Fig. 6 Proposed working model for the regulation of OsASTOL1 in sulfur metabolism in rice.** Schematic overview of enzymatic reactions (black arrows) and metabolites involved in the sulfur metabolism and its regulatory mechanism. In wild type rice, OAS is produced by cysteine synthase complex (CSC), which results in two effects: (i) induces expression of genes in sulfate uptake and assimilation (blue arrow) to enhance the supply of sulfide; (ii) dissociates the cysteine synthase complex (solid red line) to inactivate SAT. These effects control the flux of S towards cysteine synthesis in vivo. However, in the *astol1* mutant, the mutated OsASTOL1$^{S189N}$ protein is difficult to be dissociated from the CSC by OAS (dotted red line), which results in SAT being constitutively active and producing more OAS. Increased OAS level enhances sulfate/selenate uptake and assimilatory reduction and arsenic detoxification (bigger size of black and blue arrows).

contrast to OAS biosynthesis, the S-precursor for cysteine biosynthesis is exclusively generated in plastids in a photosynthesis-dependent manner[21]. The uncoupling of the OAS-sensing function of OAS-TL from its regulatory role inside the CSC stimulated sulfate uptake from the soil and its reductive assimilation in plastids significantly. It remains to be shown if the CSCs in other subcellular compartments contribute to the coordination of S-flux into cysteine with translation demand or have distinct features (e.g., coordination with supply of other macronutrients, or coordination with stress responses). The OsASTOL1$^{S189N}$ mutation provides a versatile tool to dissect these potential regulatory functions of the individual CSCs in each of these compartments by allowing specific engineering of CSCs at the subcellular level.

In conclusion, our study demonstrates that S assimilation can be effectively controlled by engineering the sensitivity of the CSC against dissociation by OAS, highlighting the physiological importance of the regulatory function of the CSC. Triggering of S assimilation in rice has multiple beneficial impacts on the food security (sustainable growth) and the nutritional value of rice since it enhances the tolerance of rice towards arsenic and heavy metals, and positively affects Se content but limits As content of grains. The key to optimization of the beneficial traits is to adjust the balance between the wild type like and the OsASTOL1$^{S189N}$-like OAS-TL protein, as too much-mutated protein had a deleterious effect on plant growth possibly through disturbance in the balance between C, N, and S metabolism. This goal can be achieved by expression of wild-type *OsASTOL1* in the *astol1* (+/+) mutant as demonstrated in our study (Fig. 5), or by the expression of mutant *Osastol1* in wild-type background driven by a suitable promoter.

## Methods

**Plant materials and growth conditions**. Rice (*Oryza sativa* cv. Kasalath) seeds were mutagenized by treating them with 0.8% ethyl methanesulfonate (EMS) for 12 h. $M_2$ seeds obtained from self-pollinated $M_1$ plants were used for the screening

of arsenite (As(III)) tolerant mutants based on root elongation at 20 µM As(III). The $M_3$ generation of the candidate *astol1* mutant was used to verify the As(III) tolerant phenotype. In brief, the root length of the seedlings after germination was recorded as $R_0$. Seedlings were transferred to deionized water for 24 h and the root length was recorded as $R_1$. Seedlings were then transferred to deionized water or deionized water containing 20 µM As(III) for 48 h and root length was recorded as $R_2$. The relative root elongation (RRE) was calculated as $(R_2-R_1)/(R_1-R_0) × 100$. Similarly, the RRE of *astol1* was calculated when treated with 8 µM arsenate (As(V)).

For hydroponic experiments, rice plants were grown in ½ Kimura B nutrient solution in a growth chamber with 65% relative humidity under 12 h/12 h light/dark (light intensity ~ 300 µmol m$^{-2}$ s$^{-1}$) cycle at 28 °C/25 °C day/night temperatures. The composition of half-strength Kimura B nutrient solution was: 0.09 mM KH$_2$PO$_4$, 0.27 mM MgSO$_4$, 0.18 mM (NH$_4$)$_2$SO$_4$, 0.09 mM KNO$_3$, 0.18 mM Ca(NO$_3$)$_2$, 3 µM H$_3$BO$_3$, 0.5 µM MnCl$_2$, 1 µM (NH$_4$)$_6$Mo$_7$O$_{24}$, 0.4 µM ZnSO$_4$, 0.2 µM CuSO$_4$, and 20 µM Fe(III)-EDTA. The nutrient solution was renewed every 3 days. For field experiments, plants were grown in paddy fields in the experimental farm of Nanjing Agricultural University in Nanjing (Jiangsu province, June-October; soil As content 12.5 mg kg$^{-1}$ and soil Se content 0.61 mg kg$^{-1}$) or Lingshui (Hainan province, January-May; soil As content 3.1 mg kg$^{-1}$ and soil Se content 0.11 mg kg$^{-1}$).

**MutMap-based gene cloning**. To clone the causal gene responsible for the *astol1* mutant phenotype, the *astol1* mutant as paternal was backcrossed with wild type (Kasalath) to generate $F_2$ progeny. The $F_2$ plants which showed tolerance to 20 µM As(III) and the early leaf death phenotype were selected for DNA extraction for whole-genome resequencing using the MutMap method[32]. In brief, DNA was extracted from 42 $F_2$ plants showing the As(III)-tolerant and early leaf death phenotype and 10 wild-type plants (Kasalath) and mixed in an equal ratio to prepare pooled genomic DNA of *astol1* mutant and wild-type, respectively. Each of the two pooled DNA samples was used for the preparation of libraries for Illumina sequencing according to the protocol for the Paired-End DNA Sample Prep kit (Illumina). The DNA library of each pool was subjected to the whole-genome sequencing using an Illumina Hiseq4000 sequencer (100× and 30× coverage for *astol1* and wild-type, respectively) and 150-bp paired-end reads were generated. After removing the adapter sequences and low-quality reads, clean reads were mapped to the reference genome sequence (Os-Nipponbare-Reference-RGAP7, MSU) using BWA software[58], followed by SNP-calling using GATK software[59]. Causative variants were calculated using the SNP-index method[32] and candidate SNPs were identified. The candidate SNPs were confirmed by PCR sequencing. The primers used are listed in Supplementary Table 1.

**Plasmid construction and plant transformation**. To generate the *pUbi-OsASTOL1* and *pUbi-Osastol1* constructs, the open reading frames (ORFs) of *OsASTOL1* and *Osastol1* were amplified from cDNA of cv. Kasalath and *astol1* homozygous mutant, respectively, and then cloned into the *Kpn*I/*Spe*I sites of the pTCK303 vector.

To generate a *pCRISPR-OsASTOL1* gene-editing construct with a single target site, a 20 bp target sequence from the second exon of *OsASTOL1* (5′-AATGGGTGAGACCATCGCCA-3′) was designed and then cloned into pOs-Cas9 vector according to Sun et al.[60].

To generate a *pCRISPR-OsASTOL1* gene-editing construct with double target sites, specific target sequences of *OsASTOL1* were blasted by CRISPR-GE (http://skl.scau.edu.cn/home/) and two target sequences (5′-AATGGGTGAGACCATCGCCA-3′ and 5′-TTGGCTCAATCAGCACACTC-3′) were selected and cloned into single guide RNA (sgRNA) expression cassettes by overlapping PCR, producing *pU3-OsASTOL1T1-sgRNA* and *pU6a-OsASTOL1T2-sgRNA* fragments, respectively. The two fragments were sequentially cloned into the *Bsa*I site of the pYLCRISPR/Cas9-MH vector.

To generate a *pOsASTOL1-GUS* construct, a 2.7-kb DNA fragment of the *OsASTOL1* upstream sequence before ATG was PCR amplified from the Kasalath genomic DNA and subcloned into the *Pac*I/*Asc*I sites of pS1aG-3 vector[61] to replace the original *HvPht1;1* promoter before the *GUS* (β-glucuronidase) reporter gene.

To generate the *OsASTOL1-YFP* and *Osastol1-YFP* constructs, the ORFs of *OsASTOL1* and *Osastol1* without the stop codon were cloned into the pEarleyGate 101(C-YFP-HA) vector using Gateway technology. In brief, the ORFs of *OsASTOL1* and *Osastol1* were PCR-amplified using *att*B adapter-containing primers and then the fragments were cloned into pDONR vector using BP Clonase (Invitrogen). *OsASTOL1-pDONR* and *Osastol1-pDONR* were subcloned into pEarleyGate 101(C-YFP-HA) vector using LR Clonase (Invitrogen).

Because the full-length OsASTOL1 protein was expressed with a very low efficiency in *E. coli* likely due to the presence of the signal peptide in the N terminus, the signal peptide (87 amino acid residues) of OsASTOL1 was deleted before expression in *E. coli*. To generate the OsASTOL1-His, Osastol1-His, AtOAS-TL A-His and AtOAS-TL A$^{S102N}$-His for in vitro enzyme assays and for protein binding analysis, the ORFs of *OsASTOL1* without the first 261 bp sequence in the N terminus, *Osastol1* without the first 261 bp sequence in the N terminus, *AtOAS-TL A* and *AtOAS-TL A$^{S102N}$* (site-directed mutation *AtOAS-TL A*) without the stop codon were amplified and cloned into the *Nde*I/*Xho*I sites of pET-29a vector, respectively. Constructs of different alleles at the 189th site of *OsASTOL1-His* were generated by site-directed mutagenesis.

For in vitro pull-down assays, the construct of *pET-28a-His-AtSAT5* was used[62]. To generate the *OsASTOL1*, *Osastol1*, *AtOAS-TL A* and *AtOAS-TL A$^{S102N}$* constructs, the ORFs of *OsASTOL1* (without the first 261 bp sequence in the N terminus), *Osastol1* (without the first 261 bp sequence in the N terminus), *AtOAS-TL A*, and *AtOAS-TL A$^{S102N}$* with the stop codon were PCR-amplified and cloned into the *Nde*I/*Sac*I sites of pET-32a vector, respectively. To generate the *S-tag-OsASTOL1*, *S-tag-Osastol1*, *S-tag-AtOAS-TL A*, and *S-tag-AtOAS-TL A$^{S102N}$* constructs, the ORFs of *OsASTOL1* (without the first 261 bp sequence in the N terminus), *Osastol1* (without the first 261 bp sequence in the N terminus), *AtOAS-TL A*, and *AtOAS-TL A$^{S102N}$* with the stop codon were PCR-amplified and cloned into the *Nco*I/*Sac*I sites of pET-29a vector, respectively.

All constructs were verified by DNA sequencing analysis. The primers used are listed in Supplementary Table 1. *pTCK303-Ubi-OsASTOL1* (for overexpression), *pOs-Cas9-OsASTOL1* (for knock-out), and *pTCK303-Ubi-Osastol1* (for complementation test) were transformed into wild-type Kasalath. *pTCK303-Ubi-OsASTOL1* and *pYLCRISPR/Cas9-MH-OsASTOL1* were transformed into *Osastol1* heterozygous mutant for complementation test. *pOsASTOL1-GUS* vector was transformed into cv. Nipponbare for tissue localization assay. All plasmids were transformed into rice plants using *Agrobacterium*-mediated transformation[63].

**Hydroponic experiments**. For uptake experiments, 4-week-old rice seedlings of wild-type Kasalath and *astol1* heterozygous mutant were exposed to half strength Kimura B nutrient solution containing 5 μM As(III), 5 μM As(V), or 2 μM Se(VI) for 3 days.

For complementation test, 5-week-old rice seedlings of wild-type Kasalath, *astol1* heterozygous and homozygous mutants, knockout lines in different background genotypes were exposed to half strength Kimura B nutrient solution containing 5 μM As(III) for 3 days. The background genotypes of rice plants were identified by a derived cleaved amplified polymorphic sequence (dCAPS) marker. The PCR products were digested with *Dde*I. The primers used are listed in Supplementary Table 1.

To evaluate the effects of *OsASTOL1* overexpression or knockout, 5-week-old rice seedlings of wild-type Kasalath, *astol1* heterozygous mutant, knockout lines, overexpression lines, and null lines in different background genotypes were exposed to half strength Kimura B nutrient solution containing 5 μM As(III) for 3 days. The background of transgenic plants was genotyped by a derived cleaved amplified polymorphic sequence (dCAPS) marker (*Dde*I) targeting the mutated site of *Osastol1* in *astol1*. The primers used are listed in Supplementary Table 1.

**Elemental analysis**. Roots and shoots of rice plants in hydroponic experiments were separated and sampled at the end of the treatments. Plants grown in field

experiments were separated into different tissues. Plant samples were dried at 65 °C for 3 days, ground to fine powder and then digested with HClO$_4$/HNO$_3$ (15/85, v/v) in a heating block. Brown rice samples were digested with high-purity HNO$_3$ in a microwave oven. The concentrations of As, Se, and other mineral elements were determined using inductively coupled plasma mass spectrometry (ICP-MS, Nex-ION 300X, PerkinElmer). Blanks and certified reference materials (rice flour NIST1568b from National Institute of Standards and Technology, Gaithersburg, MD, USA and spinach leaves GBW10015 from Institute of Geophysics and Geochemical Exploration, Langfang, China) were included for quality control in the analysis.

**Analysis of metabolites**. Non-protein thiols (glutathione and phytochelatins) were quantified using a HPLC-Fluorescence method according to Minocha et al.[64]. In brief, rice roots were homogenized with a thiol extraction solution (0.1% trifluoroacetic acid (TFA), 6.3 mM diethylenetriamine-pentacetate acid (DTPA), 10 mM tris(2-carboxyethyl)phosphine (TCEP)). After centrifugation, the supernatant was derivatized with monobromobimane (mBBr). The thiols were separated by reversed-phase HPLC and quantified with standard curve by collecting the fluorescence signal under 380 nm excitation and 470 nm emission.

Metabolites involved in sulfur metabolism were analyzed by Metware Biotech Co., Ltd (Wuhan, China). In brief, roots and shoots of 5-week-old wild-type rice and *astol1* heterozygous mutant were frozen in liquid nitrogen and freeze dried. Three biological replicates were included for each line with six plants for each biological replicate. The metabolites from freeze-dried samples were extracted overnight by 80% ethanol and qualitatively analyzed using a liquid chromatography-electrospray ionization-tandem mass spectrometry (LC-ESI-MS/MS) system according to Chen et al.[65]. The metabolites were quantified using standard curves. For the measurement of sulfate, roots and shoots of rice plants were extracted with deionized water at 4 °C for 1 h and heated at 95 °C for 15 min. After centrifugation and purified through 0.22-μm Millipore filters, the supernatant was quantified on a Dionex ICS-600 ion chromatography system.

**RNA extraction and quantitative real-time PCR**. Total RNA was extracted from different plant tissues using a RNeasy plant extraction kit (Bioteke) following the manufacturer's protocol. First-strand cDNA was synthesized using 2 μg RNA using oligo dT(23) primer according to the manufacturer's instructions of the HiScript Q Select RT SuperMix for quantitative PCR kit (Vazyme). Quantitative real-time PCR was performed on a CFX96 Real-Time PCR Detection System (Bio-Rad) using an SYBR Green Master Mix kit (Vazyme) according to the manufacturer's instructions. The rice *OsHistone H3* gene was used as the internal reference. The expression level of each gene was calculated as $2^{-\Delta Ct}$ relative to the internal reference. The primers used are listed in Supplementary Table 1.

**Tissue and subcellular localization of OsASTOL1**. To investigate the expression pattern of *OsASTOL1*, the GUS reporter activity was analyzed by histochemical staining. Roots and leaves of *pOsASTOL1-GUS* transgenic plants at the seedling stage were collected and incubated with a GUS staining solution for 2 h at 37 °C. Leaves were decolorized by 90% acetone overnight. The expression levels of *OsASTOL1* in the roots and leaves of cv. Kasalath and in various tissues/organs of cv. Nipponbare at different growth stages were quantified by quantitative real-time PCR. The samples were collected according to the method described by Sun et al.[60].

To investigate the subcellular localization of OsASTOL1, the *pEarleyGate 101-OsASTOL1-YFP* and *pEarleyGate 101-Osastol1-YFP* constructs were transformed into rice protoplasts using polyethylene glycol (PEG)-mediated transformation[61], or in tobacco (*Nicotiana benthamiana* Domin) leaf epidermal cells using *Agrobacterium*-mediated transformation[66]. The pEarleyGate 104 (N-YFP) vector containing a 35 S::YFP construct was used as a control. Protoplasts isolated from rice or tobacco were observed using a confocal laser scanning microscope (Zeiss). YFP fluorescence was observed at 525 nm for emission and 510 nm for excitation, and chlorophyll autofluorescence was detected at 575–640 nm for emission and at 546 nm for excitation.

**Immunoblotting analysis**. For immunological detection of OsASTOL1 in rice, total soluble proteins were extracted from 200 mg leaf sample with 1 ml plant protein extraction buffer (CW0885, CWBIO). Proteins were denatured by heating for 10 min in a boiling-water bath and separated on 12% SDS-PAGE. Subsequently, proteins were transferred onto a polyvinylidene difluoride (PVDF) membrane and immunoblots were probed with the following antibodies in the TBST solution (100 mM Tris-HCl, 150 mM NaCl, 0.1%(v/v) Tween 20, pH 7.4) with 5% nonfat dry milk. A rabbit anti-OAS-TL A antibody (against full-length Arabidopsis cytosolic OAS-TL A, 1:5000 dilution) and a rabbit anti-RbcL antibody (AS03037, Agrisera, 1:10000 dilution) were used as primary antibodies and HPR-conjugated goat anti-rabbit IgG (AT0097, CMCTAG, 1:10000 dilution) was used as secondary antibody. Chemiluminescence was performed using ECL Western Blotting Substrate (IF6747, CMCTAG) according to manufacturer's instructions.

To produce different variants of AtOAS-TL A and OsASTOL1, the plasmids *pET-29a-AtOAS-TLA-His*, *pET-29a-AtOAS-TLA$^{S102N}$-His*, *pET-29a-OsASTOL1-His*, *pET-29a-Osastol1-His*, and different allele of *pET-29a-OsASTOL1-His* were transformed into *E. coli* strain BL21 (DE3). The bacterial cells were cultured at

37 °C until the $OD_{600}$ reached 0.5, and then induced by 0.3 mM isopropyl-β-D-thiogalactopyranoside (IPTG) at 16 °C for 16 h. Cells were collected and sonicated by an ultrasonic cell crusher. After centrifugation and heating, denatured proteins in the supernatant were separated by 12% SDS-PAGE. Western blot analysis was performed using a mouse anti-His-tag antibody (AF5060, Beyotime,1:2000 dilution). HPR-conjugated goat anti-mouse IgG (AT0098, CMCTAG, 1:10000 dilution) was used as the secondary antibody.

**Analysis of OAS-TL and SAT enzyme activity**. To determine the OAS-TL enzyme activity, total soluble proteins were extracted from 200 mg rice leaves with 1 ml plant protein extraction buffer (CW0885, CWBIO). Recombinant OAS-TL proteins were expressed in *E. coli* after induction by 0.3 mM isopropyl-β-D-thiogalactopyranoside (IPTG) at 16 °C for 16 h. Cells were sonicated and filtered through 0.45-μm Millipore filters. The supernatant (crude extract protein) was added to Ni-NTA packed columns (Sangon Biotech Co. Ltd, Shanghai) and recombinant proteins were eluted with a PBS buffer (20 mM $Na_3PO_4$, 100 mM NaCl, pH 7.4) containing 150 mM imidazole. All proteins were quantified with a BCA Protein Assay Kit (CW0014S, CWBIO). The OAS-TL enzyme activity was determined at 25 °C in a total volume of 100 μl containing 50 mM HEPES/KOH (pH 7.5), 5 mM $Na_2S$, 10 mM OAS, 5 mM DTT and 1-2 μl of crude extract proteins according to Heeg et al.[25]. Recombinant proteins were diluted 1:100 before the enzyme activity assay.

To determine the SAT enzyme activity, total soluble proteins were extracted from 200 mg rice leaves with 1 ml plant protein extraction buffer (CW0885, CWBIO). The SAT activity[53] was assayed as described before, but with some modification. The SAT activity was first assayed at 25 °C in a total volume of 100 μl containing 50 mM HEPES/KOH (pH 7.5), 10 mM serine, 1 mM acetyl-CoA, 5 mM DTT and 70 μl protein extract. After reaction for 30 min, 50 μl 20% trichloroacetic acid (TCA) was added and the content was centrifuged. The pH of the supernatant was adjusted to 7.4 with 40 μl 2 M Tris, to which 50 μl 50 mM $Na_2S$ and 4 U purified AtOAS-TL A were added to ensure an excess of OAS-TL activity. The reaction mixture was incubated at 25 °C for 20 min to allow OAS produced by SAT to be transformed to cysteine. The production of cysteine was determined with an acid-ninhydrin solution.

**Functional analysis of OAS-TL in *E. coli* NK3 strain**. To investigate whether OsASTOL1 functions as an OAS-TL enzyme, the empty vector *pET-29a*, *pET-29a-OsASTOL1-His*, and *pET-29a-Osastol1-His* were transformed into the cysteine auxotroph *E. coli* NK3 strain. *pET-29a-AtOAS-TL A-His* was used as a positive control. All bacterial cells were grown on M9 minimal medium (0.2 g $L^{-1}$ leucine, 0.2 g $L^{-1}$ tryptophan, 50 μg $ml^{-1}$ kanamycin, 1 mM IPTG, and 1.5% agar) with or without 0.5 mM cysteine.

**Size exclusion chromatography analysis**. Untagged OsASTOL1 and OsAS-TOL1[S189N] were expressed and purified according to Wirtz et al.[26]. Molecular weights of OsASTOL1 and OsASTOL1[S189N] were determined by size exclusion chromatography (SEC) using a FPLC[TM] system. Briefly, 1 ml of OsASTOL1 or OsAS-TOL1[S189N] (1 mg) was injected into the sample loop and was eluted through the HiLoad[TM] 16/60 Superdex 200 prep grade column (120 ml, GE Healthcare) in buffer (20 mM Tris-HCl, 150 mM NaCl, pH 7.5) at 1 ml/min flow rate. Calibration of the Superdex 200 column was performed with the high and low molecular weight gel filtration calibration kits (GE Healthcare) according to the manufacturer's instructions. The molecular weights of OsASTOL1 and OsASTOL1[S189N] were calculated from the calibration curve with the measured elution volumes.

**Sequence alignment and visualization of protein structure**. Multiple sequence alignments of OAS-TL proteins were conducted using the ClustalW method in DNAMAN software. Visualization of protein structure and site-directed mutation of *Arabidopsis* cytosol OAS-TL A (PDB ID: 1z7w) were performed by PyMOL software.

**In vitro pull-down assay**. To investigate the interactions of OsASTOL1, OsAS-TOL1[S189N], AtOAS-TL A or AtOAS-TL A[S102N] with AtSAT5 and the dissociation effect by OAS, a revised in vitro pull-down assay was conducted. In brief, soluble proteins were extracted from cells harboring the *pET-28a-His-AtSAT5* plasmid and added to a $NiCl_2$ containing HiTrap[TM] Chelating column (GE Healthcare). The column was washed with Buffer W (50 mM Tris-HCl, pH 8.0, 250 mM NaCl, 80 mM Imidazol), followed by washing with Buffer W containing 10 mM OAS (Solar Bio) to remove the bacterial OAS-TL. The crude extract containing untagged plant OAS-TL proteins was added to the above column containing the His-tagged AtSAT5 protein. The untagged OAS-TL protein was specifically eluted with Buffer W containing 10 mM OAS. After washing with Buffer W, the His-tagged AtSAT5 was eluted with Buffer E (50 mM Tris-HCl, pH 8.0, 250 mM NaCl, 400 mM Imidazol). The fractions obtained from different steps of elution were analyzed by SDS-PAGE followed by Coomassie staining.

For immunological detection of OAS-TL and SAT proteins, the His-AtSAT5 and S-tagged OAS-TL proteins were used to perform the same steps as described above but with a different Ni-NTA packed columns (Sangon Biotech Co. Ltd, Shanghai) and a new washing buffer (50 mM Tris-HCl, pH 8.0, 250 mM NaCl, 20 mM Imidazol). A rabbit anti-S-tag antibody (101290-T38, Sino Bio, 1:5000 dilution) and a mouse anti-His-tag antibody (AF5060, Beyotime,1:2000 dilution) were used as the primary antibodies.

**Microscale thermophoresis analysis**. Ten C-terminal residues[38] of AtSAT5 (AT5G56760, sequence: FISEWSDYII), required for binding with OAS-TL, was synthesized by GenScript Biotech (Nanjing). Recombinant His-tagged OAS-TL proteins were labeled with RED-tris-NTA 2nd Generation Dye for 30 min. For the binding assay, 100 μM labeled His-tagged OAS-TL protein was incubated with a serial dilution of the ligand (OAS or AtSAT5C10 peptide) in the PBS buffer (pH 7.6) containing 0.05% Tween 20. Microscale thermophoresis (MST) analysis was performed on a Monolith NT.115 instrument (Nano Temper, Germany) using auto-detected LED power (5–90%) and MST power (Medium or High). The data were evaluated using the MO Affinity Analysis software.

**Reporting summary**. Further information on research design is available in the Nature Research Reporting Summary linked to this article.

## Data availability
Data supporting the findings of this work are available within the paper and its Supplementary Information files. A reporting summary for this article is available as a Supplementary Information file. The genetic materials generated and analyzed during the current study are available from the corresponding author upon request. The source data underlying Fig. 1b, d, g–i, 2b–e, 3, 4b–g, 5, as well as Supplementary Figs. 1a, 2b–d, 3b, 4–6d–f, 10a, 12b, 14, 15b–e, 16–20d–l, 21c–e, and Supplementary Table 4 are provided as a Source Data file.

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

## Acknowledgements

This work was supported by the grants from the Natural Science Foundation of China (31520103914), the Ministry of Science and Technology Key R&D program (2016YFD0100704), the Innovative Research Team Development Plan of the Ministry of Education of China (grant no. IRT_17R56), the Fundamental Research Funds for the Central Universities (grant no. KYT201802) and German Research Foundation (DFG, WI3560/1-2, WI3560/6-1 and HE1848/15-2, HE1848/19-1).

## Author contributions

F.J.Z. and S.K.S designed the research. S.K.S., X.X., Zhong T., and Zhu T. performed experiments. S.K.S., X.Y.H., and M.W. analyzed data. S.K.S., F.J.Z., M.W., and R.H. wrote and revised the manuscript with inputs of other authors.

## Competing interests

The authors declare no competing interests.
