## [Peer Review File · Nature Communications]

REVIEWER COMMENTS

Reviewer #1 (Remarks to the Author):

Reducing As level in rice is an important issue for agriculture because rice is a major source of dietary intake of As in human. Authors isolated a rice mutant showing arsenite tolerance (astol1) and identified a point mutation in the gene encoding O-acetylserine (thiol)lyase (OAS-TL) that is an enzyme catalyzing the reaction of inorganic sulphide with O-acetylserine (OAS) to form the cysteine (Cys). The point mutation led to Ser189Asn mutation and astol1 mutant lost the OAS-TL activity. However, Ser189Asn did not affect the formation of cysteine synthase complex (CSC) consisting of serine acetyltransferase (SAT) and OAS-TL and the SAT activity of astol1 mutant rather increased than that of WT because OAS-induced dissociating effect of CSC was inhibited due to low affinity of OAS-TL carrying Ser189Asn to OAS. The increased SAT activity in astol1 contributed to the increasing levels of OAS, Cys, GSH, PCs etc., resulting in the increased S and Se accumulation and the decreased As level in rice grains. The present data are interesting and novel, and the experiments are well designed to reveal the complex phenotypes of low As and high S and Se in astol1 mutant and this manuscript includes strong evidences. However, several questions and criticisms, especially for the phenotypes of CRISPR plants and OX-rice lines, are raised by me. Authors should meet my concerns described below.

1. Lines 92-102: Why is increasing light intensity improved the growth of heterozygous astol1 (+/-)?
2. Lines 115-116: SNP-index graphs of chromosomes 1 to 12 should be given in Suppl.
3. Lines 127-142: Because Osastol1 protein shows a loss of function, the mutant must be a knockout line of osastol1 gene. But authors further knocked out the osastol1 gene in the mutant by CRISPR. I am not sure about the objective of this experiment. In addition, I am curious why all CRISPR lines in even at astol1(+/+) background was completely recovered the astol1(+/+) mutant-induced phenotypes in growth and As concentration. Do authors want to say that other mutations except S189N in OsASTOL1 do not induce phenotypes observed in osastol1 mutant?
4. Lines 274-293: I understand that Ox-rice lines overexpressing WT-OsASTOL1 in astol1(+/+) astol1(+/-) backgrounds can improve their growth because astol1 mutation inhibited the growth. But I am very curious why the phenotypes of As, S, and Se in Ox-lines did not show the similar phenotypes with WT irrespective of overexpression of WT-OsASTOL1. Authors addressed that growth inhibition of astol1(+/+) can be overcome by balancing the mutant and WT OsASTOL1 protein. Why are As phenotypes no balancing? Please discuss in more detail.
5. Lines 284-287: Straw As concentrations also should be shown in Fig. 5. In young plants, astol1 accumulated more As in the roots than WT, but was not difference in shoot As concentration as compared to WT. If straw As concentration shows the similar trend with young plants, I am interesting in the As distribution pattern of the mutant because nodes play an important role in controlling the grain As. The astol1 mutant may have higher amounts of PCs in nodes and As concentration than the WT. If authors have some data, should be shown.
5. Lines 321-323: Is there any direct evidence about catalysis of other isoforms of OAS-TL or data suggested?
6. I think that I could understand the molecular regulation of phenotypes in respect with As and Se in astol1 mutant. But I am still questionable why the growth of homozygous astol1 (+/+) mutant was inhibited significantly, although the rice growth did not affect by disturbance of OAS-TL activity. Do you think that the growth inhibition of astol1 mutant is caused by restless SAT activity?

Reviewer #2 (Remarks to the Author):

In this study, authors isolated a new mutant tolerant to arsenite and identified a new amino acid important for the activity of OAS-TL and for constructing the complex named CSC. By analyzing

the o-acetyl-L-serine-induced dissociation of CSC, authors beautifully probed the importance of serine 189. Also, the authors demonstrated the higher accumulation of S and Se in seed grain and discussed the activation of sulfur acquisition and assimilation in this mutant. However, there are some missing pieces to lead the conclusion as pointed below.

1. The mutation affected the activity of OAS-TL, there are other possibilities such as this mutation affects dimer construction between OAS-TL. Did the authors compared the native MW by doing gel chromatography? Also, I'm wondering how the substitution to threonine show 50% of the activity to the wild-type protein. Did the authors analyze the phosphorylation of this OAS-TL with or without OAS? The different mobility in SDS-PAGE also suggested that this base substitution influenced the protein modification.

2. The authors need to demonstrate the role of this OAS-TL under the normal condition. The secondary structure of this isoform is similar to the cytosolic isoform of Arabidopsis but localized to the plastid. Is it the only OAS-TL existing in rice plastid? In the case of Arabidopsis, there is a feedback inhibition for cytosolic OAS-TL activity, and the corresponding amino acid was demonstrated. Is the amino acid is conserved in this isoform? Because this is the semi-dominant mutation, there is a possibility to affect the activity of other OAS-TL by comprising the heterosexual dimer if it is existing in the plastid. Also, the native ASTOL1 seems localized to both plastid and cytosol comparing the mutant exactly localizing to the plastid in Fig 2 and Fig S10.

3. The expression of this gene seems higher in the stem and sheath. As accumulation in the grain is determined by the chelation and sequestration around the node, authors need to analyze the As and thiol levels in the other tissues than the grain or the seedlings.

4. Arsenite is known to be chelated by phytochelatin rather than GSH mostly in the node. The authors only presented the PC level in roots, but need to analyze PC levels in the grain and the different part of the plant. Understandably, sulfur accumulation in seedling is increased by the increased activity of sulfate transporter, but, strangely, seed S content in seeds is increased like this. I think the authors should discuss this point. Also, when the plastid OAS-TL is mutated, how the S assimilation can be increased? It is not a matter of sulfate uptake, but the matter of reduction and assimilation. I think the authors need to analyze the activity of whole OAS-TL activity in plants, if possible by separating the cytosol and plastid.

5. In Fig. 4, the authors show the ratio between the control condition and arsenite application, but the difference between the two genotypes is also important. Supplemental Table showing the metabolite levels would be requested. Also, I'm wondering why cystathionine content is so much affected rather than GSH.

Minor points

Fig 1g in L85 should be Fig 1h.

The order of Supplemental Fig 8-10 should be changed to the order of their appearance in the text. Fig 4a is not cited in the text.

Reviewer #3 (Remarks to the Author):

This is a very interesting work that identifies the molecular basis of the rice arsenic tolerance mutant *astol1*. Interestingly the mutation confers an increased stability of the Cysteine Synthase Complex (CSC) due to a point mutation on the OASTL rice protein. The mutation also affects OASTL enzyme activity leading to increased OAS accumulation. Based on the important regulatory role of OAS in cysteine biosynthesis the mutant displays increased sulphur, glutathion and

phytochelatin content that leads to increased arsenic accumulation in roots preventing its accumulation in grains. In addition, the mutant also increased Selenium accumulation. Therefore, uncovering the molecular mechanisms underlying astol1 mutation, provides a novel strategy to obtain plants with less accumulation of arsenic in the aerial part together with increased selenium accumulation.

I believe the quality of the results obtained are excellent, however I would like to add some comments regarding the way in which the work is introduced and described.

I think the CSC complex and functionality should be further explained in the introduction. Previous work regarding this issue should be mentioned. Indeed, it has been shown that 80% of the SAT activity comes from the mitochondrial isoform while OASTL activity comes preferentially from the plastidial and cytosolic isoforms. Therefore, the role of CSC in each compartment is currently controversial and that should be further discussed in the context of the obtained data. In addition, both OASTL and SAT genes are not transcriptionally regulated but exposure to stress conditions can induce transcript accumulation (Lehmann et al 2009. Mol Plant 2:390-406). Therefore, quantification of transcript abundance in response to As(III) in wild-type and mutant plants would be desirable and the results discussed under the context of previous data.

The importance of arsenic in rice natural variation should be explored. I would suggest to look for the astol1 SNP or analogous in the sequence database of different rice accessions. It would be particularly relevant if the arsenic tolerance of the accessions were already determined, which would further support the biological importance of the mutation uncovered here and therefore improve the quality of the results presented.

Finally, I would further discuss the possible reasons that lead to the pleiotropic effects of the mutation, in particular the effect of light intensity.

In conclusion, I think this is an excellent work providing a new perspective to obtain rice plants with better performance in arsenic contaminated soils preventing its accumulation in grains and increased selenium resulting in fortified crops. The results are novel, the conclusions are very well sustained and the methods are sufficiently explained.

Minor comments:

- Figure 1g cited in line 85 should be Figure 1h.
- In line 150 it must be said that astol1 point mutation does not affect the OASTL chloroplastic localization in rice
- The complementation of NK3 complementation assay should be explained with further detail.
- The use of the OASTL cytosolic isoform from Arabidopsis should be further justified.

Antonio Leyva

Responses to Reviewers' comments

Reviewer #1 (Remarks to the Author):

Reducing As level in rice is an important issue for agriculture because rice is a major source of dietary intake of As in human. Authors isolated a rice mutant showing arsenite tolerance (*astoll*) and identified a point mutation in the gene encoding O-acetylserine (thiol)lyase (OAS-TL) that is an enzyme catalyzing the reaction of inorganic sulphide with O-acetylserine (OAS) to form the cysteine (Cys). The point mutation led to Ser189Asn mutation and *astoll* mutant lost the OAS-TL activity. However, Ser189Asn did not affect the formation of cysteine synthase complex (CSC) consisting of serine acetyltransferase (SAT) and OAS-TL and the SAT activity of *astoll* mutant rather increased than that of WT because OAS-induced dissociating effect of CSC was inhibited due to low affinity of OAS-TL carrying Ser189Asn to OAS. The increased SAT activity in *astoll* contributed to the increasing levels of OAS, Cys, GSH, PCs etc., resulting in the increased S and Se accumulation and the decreased As level in rice grains. The present data are interesting and novel, and the experiments are well designed to reveal the complex phenotypes of low As and high S and Se in *astoll* mutant and this manuscript includes strong evidences.

Response: We thank the reviewer for the very positive comments on our work.

However, several questions and criticisms, especially for the phenotypes of CRISPR plants and OX-rice lines, are raised by me. Authors should meet my concerns described below.

1. Lines 92-102: Why is increasing light intensity improved the growth of heterozygous *astoll* (+/-)?

Response: This is indeed a very interesting question. In their natural habitats, rice typically encounters intense light (on average $> 1000 \mu\text{mol m}^{-2} \text{s}^{-1}$) and is well adapted to this condition. For that reason, the “norm” light condition in open paddy fields is still in the optimal range for photosynthesis. Consequently, this light regime would allow efficient carbon incorporation to meet the demand for enhanced synthesis of OAS, Cys and other elevated thiol compounds in *astoll*(+/-) and *astoll*(+/+). Another factor that may add to wild-type like growth of *astoll*(+/-) in paddy fields is that enhanced GSH levels prime plants towards diverse environmental stresses that did not occur in low light conditions in growth chambers. This misbalanced defense response might also add to the slower growth of *astoll*(+/-) when compared to the wild type under this artificial condition. In open paddy field, the wild type

and *astol1* mutant need to adapt to fluctuating environmental cues including diverse stresses requiring GSH synthesis, e.g. adaptation to varying light regimes, pathogens and water supply (Parisy *et al.* 2007; Dominguez-Solis *et al.* 2008; Park *et al.* 2013; Müller *et al.* 2017). Under these natural occurring stresses *astol1*(+/-) performed as well as the wild type, which might be caused by the pre-adapted thiol containing defense compounds that are useless for plants grown in under low light in the lab. We have added the following sentences in the Discussion section of the revised manuscript to explain our results (**Lines 362-374**): “Increased biosynthesis of OAS, Cys and other elevated thiol compounds in the *astol1* mutants causes a higher demand for carbon. Consequently, norm light conditions in paddy fields helped the mutants to overcome the imposed challenge for enhanced synthesis of the diverse carbon and sulfur-containing metabolites. The relatively poor growth of the mutants under low light intensity in the absence of any stresses might be also partially explained by a misbalance of enhanced level of sulfur-containing defense compounds in the artificial absence of stresses, e.g., high light and pathogens, which demand sulfur-containing defense compounds. However, in open paddy fields the potentially defense-primed *astol1* (+/-) mutants performed as well as the wild type, since the wild type was also challenged with environmental cues inducing sulfur-defense metabolite synthesis.”

References:

- Parisy, V., *et al.* (2007). Identification of PAD2 as a γ -glutamylcysteine synthetase highlights the importance of glutathione in disease resistance of Arabidopsis. *The Plant Journal*, 49(1), 159-172.
- Dominguez-Solis, J. R., *et al.* (2008). A cyclophilin links redox and light signals to cysteine biosynthesis and stress responses in chloroplasts. *Proceedings of the National Academy of Sciences*, 105(42), 16386-16391.
- Park, S. W., *et al.* (2013). Cyclophilin 20-3 relays a 12-oxo-phytodienoic acid signal during stress responsive regulation of cellular redox homeostasis. *Proceedings of the National Academy of Sciences*, 110(23), 9559-9564.
- Müller, S. M., *et al.* (2017). The redox-sensitive module of cyclophilin 20-3, 2-cysteine peroxiredoxin and cysteine synthase integrates sulfur metabolism and oxylipin signaling in the high light acclimation response. *The Plant Journal*, 91(6), 995-1014.

2. Lines 115-116: SNP-index graphs of chromosomes 1 to 12 should be given in Suppl.

Response: The SNP-index results are now shown in the revised version of the manuscripts as

new **Supplementary Fig. 5**.

3. Lines 127-142: Because Osastol1 protein shows a loss of function, the mutant must be a knockout line of osastol1 gene. But authors further knocked out the osastol1 gene in the mutant by CRISPR. I am not sure about the objective of this experiment. In addition, I am curious why all CRISPR lines in even at astol1(+/+) background was completely recovered the astol1(+/+) mutant-induced phenotypes in growth and As concentration. Do authors want to say that other mutations except S189N in OsASTOL1 do not induce phenotypes observed in osastol1 mutant?

Response: We apologize for not explaining the experiment's rationale fully in the previous version of the manuscript. The reviewer is in principle right in that only S189N in OsASTOL1 induces the mutant phenotype, but our experiments provide even further information. *astol1* is a semi-dominant mutant with both a loss of function (OASTL activity of OsASTOL1) and a gain of function (increased stability of the CSC-formed by OAS-TL and SAT and therefore increased SAT activity). If the loss-of-OAS-TL activity in *astol1* mutant would have been responsible for the phenotype, the destruction of the OASTL activity of OsASTOL1 in the wild type rice by CRISPR/Cas9 would have caused the same phenotype as the *astol1* point mutation. This was not the case (please see **Supplementary Fig. 15c-e**), which is in full agreement with the characterization of the Arabidopsis loss-of-activity function *oastl* mutants (Heeg *et al.*, 2008; Watanabe *et al.*, 2008). For this reason, we claim that the gain-of-function to enhance the SAT activity in the CSC is responsible for the dominant phenotype in the mutant.

For dominant mutants, there are two ways to prove the causal gene responsible for the mutant phenotypes: 1) express the mutant protein in a WT background and observe if the mutant phenotypes appear (e.g. Horak *et al.*, 2016; Zhou *et al.*, 2013); and 2) suppress or knock out the expression of the mutant protein in mutant plant to observe whether the mutant plant returns to the WT phenotype (e.g. Weber *et al.*, 2013). We tried both approaches. The first approach produced sterile seeds, which could not be used for phenotype analysis. Incidentally, this is consistent with too much of the mutant protein is deleterious to plant growth (please see **Lines 133-135** in the manuscript). Using the second approach, we knocked out the Osastol1 protein in *astol1* heterozygote and homozygote mutants using the CRISPR-Cas9 gene editing method. All CRISPR lines in both the *astol1*(+/-) and *astol1*(+/+) backgrounds indeed showed a complete rescue of the *astol1* mutation-induced phenotypes (**Fig. 1i and Supplementary Fig. 6c-f**), demonstrating that the S189N point mutation in the

OsASTOL1 protein (Osastol1) is the cause for the *astol1* phenotypes. This second approach abolished the gain-of-function properties of the mutant protein, therefore restoring the WT phenotypes. Indeed, other mutations except S189N in OsASTOL1 do not induce phenotypes observed in *osastol1* mutant.

References:

Heeg, C., *et al.* (2008). Analysis of the *Arabidopsis* O-acetylserine(thiol)lyase gene family demonstrates compartment-specific differences in the regulation of cysteine synthesis. *The Plant Cell*, 20, 168-185.

Watanabe, M., *et al.* (2008). Physiological roles of the β -substituted alanine synthase gene family in *Arabidopsis*. *Plant physiology*, 146(1), 310-320.

Horak, H., *et al.* (2016). A dominant mutation in the HT1 kinase uncovers roles of MAP kinases and GHR1 in CO₂-Induced stomatal closure. *The Plant Cell*, 28(10), 2493-2509.

Weber, M., *et al.* (2013). A mutation in the *Arabidopsis thaliana* cell wall biosynthesis gene pectin methylesterase 3 as well as its aberrant expression cause hypersensitivity specifically to Zn. *The Plant Journal*, 76(1), 151-164.

Zhou, F., *et al.* (2013). D14-SCF(D3)-dependent degradation of D53 regulates strigolactone signalling. *Nature*, 504(7480), 406-410.

4. Lines 274-293: I understand that Ox-rice lines overexpressing WT-OsASTOL1 in *astol1(+/+)* *astol1(+/-)* backgrounds can improve their growth because *astol1* mutation inhibited the growth. But I am very curious why the phenotypes of As, S, and Se in Ox-lines did not show the similar phenotypes with WT irrespective of overexpression of WT-OsASTOL1. Authors addressed that growth inhibition of *astol1(+/+)* can be overcome by balancing the mutant and WT OsASTOL1 protein. Why are As phenotypes not balancing? Please discuss in more detail.

Response: We thank the reviewer for carefully reading the manuscript. Indeed, the previous version of the manuscript did not contain the data showing that overexpressing WT-OsASTOL1 protein in the *astol1 (+/-)* and *astol1(+/-)* background results in balancing the As levels. In the revised manuscript, we have added these data as new panel J-I in **Supplementary Fig. 20** (the original Figure presented only the growth and shoot S data). As the reviewer suggested, Root As and Shoot S levels were partially rescued to wild type levels (balancing) after overexpression of the WT-ASTOL1 protein in the *astol1(+/-)* background. This balancing is caused by competition of the WT-ASTOL1 protein and the *astol1* protein for

binding to SAT, resulting in lowered amounts of “non-dissociable CSC” formed by interaction of SAT with the *astol1* protein.

Because the mutant protein is still present in these lines, albeit in smaller ratios to the WT protein, it is expected that there will still be an effect on sulfur metabolism. This is indeed what we have observed in terms of increased shoot S concentration and increased As sequestration in the roots of these lines compared with WT (**Supplementary Fig. 20**). The results suggest that a small dose of the mutant protein can increase sulfur metabolism and produce desirable benefits without affecting growth, as discussed already in the manuscript (**Lines 375-378 and Lines 419-424**).

5. Lines 284-287: Straw As concentrations also should be shown in Fig. 5. In young plants, *astol1* accumulated more As in the roots than WT, but was not difference in shoot As concentration as compared to WT. If straw As concentration shows the similar trend with young plants, I am interesting in the As distribution pattern of the mutant because nodes play an important role in controlling the grain As. The *astol1* mutant may have higher amounts of PCs in nodes and As concentration than the WT. If authors have some data, should be shown.

Response: We followed the suggestion of Reviewer #1 and determined for As distribution in the node, internode, leaf blade and leaf sheath in the mature plants of *astol1*(+/-) and WT. The new data are presented in the new **Supplementary Fig. 19a, b**. The As concentration in the node increased in two independent field experiments at different sites (Nanjing and Lingshui), albeit this increase was statistically significant only at Lingshui site. Although rice nodes contribute to the sequestration of As (Moore *et al.*, 2014; Song *et al.*, 2014; Chen *et al.*, 2015), our previous experiment using radioactive ⁷³As isotope tracing showed that the majority (~90%) of As taken up by rice roots is sequestered in the roots (Zhao *et al.*, 2012). Therefore, lower grain As in the mutant is likely a result of increased As sequestration in the roots. We did not determine PCs in the nodes because the samples were taken previously from mature plants (i.e. senesced) grown in paddy fields and stored at room temperature, and PCs would probably have been partially degraded. However, we have performed a new experiment and measured PCs in both roots and shoots in 4-week-old plants, in response to the comment by Reviewer #2 question 4. These new results (**Fig. 4f, g**) show that the concentrations of PCs in the mutant shoots were also increased compared to WT (**Lines 287-292**).

References:

Moore, K. L., *et al.* (2014). Combined NanoSIMS and synchrotron X - ray fluorescence

reveal distinct cellular and subcellular distribution patterns of trace elements in rice tissues.

New Phytologist, 201(1), 104-115.

Chen, Y., *et al.* (2015). The role of nodes in arsenic storage and distribution in rice. *Journal of Experimental Botany*, 66(13), 3717-3724.

Song, W.-Y., *et al.* (2014). A rice ABC transporter, OsABCC1, reduces arsenic accumulation in the grain. *PNAS* 111, 15699-15704.

Zhao, F. J., *et al.* (2012). Arsenic translocation in rice investigated using radioactive ⁷³As tracer. *Plant and Soil*, 350, 413-420.

5. Lines 321-323: Is there any direct evidence about catalysis of other isoforms of OAS-TL or data suggested?

Response: This is a very interesting and valid question that we did not address in the manuscript due to results published in a separate study and previous findings in Arabidopsis. We have recently characterized the function of a different isoform of OAS-TL in rice, OsOASTL-A1, and show that it is a cytosolic cysteine synthase involved in Cys biosynthesis (Wang *et al.* 2020). This new publication is now cited in the revised manuscript (**Lines 53-54**). In plants, sulfate reduction occurs in the plastids, while Cys can be synthesized in the cytosol, plastids, and mitochondria (reviewed in Takahashi *et al.*, 2011). In *Arabidopsis thaliana*, cysteine synthesis is catalyzed by three major isoforms of OAS-TLs located in these subcellular compartments (Wirtz *et al.*, 2004). Null mutants for each of the three major OAS-TLs (A, B, and C) are well able to grow and develop, suggesting efficient exchange of sulfide, OAS, and cysteine between cytosol and organelles (Heeg *et al.*, 2008; Takahashi *et al.*, 2011). Remarkably, elimination of OAS-TL activity in one of these compartments did not stimulate OAS-TL protein amount in the remaining compartments or compensatory increase of extractable OAS-TL (Heeg *et al.*, 2008). This information has been added in the Introduction of the revised manuscript (**Lines 54-56**). In agreement with these data, the *astol1(+/+)* mutant displays 32% lower total OASTL activity, suggesting that ASTOL1 substantially contribute to total OAS-TL activity. However, this measurement also demonstrates that there must be other OAS-TL protein, like the cytosolic OsOASTL-A1 present in rice.

References:

Wang, C., *et al.* (2020). OsOASTL-A1 functions as a cytosolic cysteine synthase and affects arsenic tolerance in rice. *Journal of Experimental Botany*, 71, 3678–3689.

Takahashi, H., *et al.* (2011). Sulfur assimilation in photosynthetic organisms: molecular functions and regulations of transporters and assimilatory enzymes. *Annual Review of Plant Biology*, 62, 157-184.

Heeg, C., *et al.* (2008). Analysis of the *Arabidopsis* O-acetylserine(thiol)lyase gene family demonstrates compartment-specific differences in the regulation of cysteine synthesis. *The Plant Cell*, 20, 168-185.

Wirtz, M., *et al.* (2004). O-acetylserine (thiol) lyase: an enigmatic enzyme of plant cysteine biosynthesis revisited in *Arabidopsis thaliana*. *Journal of Experimental Botany*, 55, 1785-1798.

6. I think that I could understand the molecular regulation of phenotypes in respect with As and Se in *astol1* mutant. But I am still questionable why the growth of homozygous *astol1* (+/+) mutant was inhibited significantly, although the rice growth did not affect by disturbance of OAS-TL activity. Do you think that the growth inhibition of *astol1* mutant is caused by restless SAT activity?

Response: We fully agree to the reviewer's hypothesis and think along the same lines. The major consequence of the *astol1* mutation is the stimulation of SAT activity by CSC formation. Enhanced formation of OAS increased sulfur metabolism and, consequently, resulted in higher cysteine and glutathione levels. Cysteine and glutathione synthesis are known to be tightly controlled in the plastids in order to maintain the redox homeostasis in plastids and cytosol (Speiser *et al.* 2018). Perturbation of the redox-homeostasis by *astol1*-induced cysteine synthesis is likely contributing to the observed growth-retarded phenotype. However, other scenarios are also possible. We recently identified that ABA biosynthesis can be triggered by cysteine application (Batool *et al.* 2018). Consequently, perturbation of ABA biosynthesis might add to the observed growth-retarded phenotype of the *astol1*(+/+) mutant. Since we cannot distinguish between these possibilities at the current stage of the analysis, we want to refrain from speculating about these possibilities in the discussion section. Although the growth retarded phenotype of the *astol1*(+/+) mutant is not in the focus of our study, we agree with the reviewer that it is scientifically interesting and have initiated experiments to analyze the impact of overkill CSC-formation in the model plant *Arabidopsis thaliana*. The expected results will have no impact on the As-reduction phenotype in the rice *astol1*(+/-) and thus are not in the scope of this manuscript.

References:

Speiser, A., *et al.* (2018). Sulfur partitioning between glutathione and protein synthesis determines plant growth. *Plant Physiology*, 177(3), 927-937.

Batool, S., *et al.* (2018). Sulfate is incorporated into cysteine to trigger ABA production and stomatal closure. *The Plant Cell*, 30(12), 2973-2987.

Reviewer #2 (Remarks to the Author):

In this study, authors isolated a new mutant tolerant to arsenite and identified a new amino acid important for the activity of OAS-TL and for constructing the complex named CSC. By analyzing the o-acetyl-L-serine-induced dissociation of CSC, authors beautifully probed the importance of serine 189. Also, the authors demonstrated the higher accumulation of S and Se in seed grain and discussed the activation of sulfur acquisition and assimilation in this mutant.

Response: we thank the reviewer for the very positive comments on our work.

However, there are some missing pieces to lead the conclusion as pointed below.

1. The mutation affected the activity of OAS-TL, there are other possibilities such as this mutation affects dimer construction between OAS-TL. Did the authors compared the native MW by doing gel chromatography?

Response: We did not address the dimerization of OAS-TL in the previous version of the manuscript because the astol1 mutation is not in the dimer interface, which is conserved among all OAS-TLs analyzed so far (Burkhard *et al.*, 1998; Bonner *et al.*, 2005). However, we agree with the reviewer that even point mutations far away from the dimer interface might have an impact on OAS-TL dimerization. For that reason, we have determined the molecular weight (stokes radius) of the wild-type OsASTOL1 and the mutated Osastol1 protein by Size Exclusion Chromatography (SEC). As expected, both proteins eluted at the same retention volume (79.2 ml), which corresponds to a molecular size of 66.9 kDa, representing the OAS-TL dimer (calculated MW of an OsASTOL1 monomer is 33.8 kDa). These results have been added as new **Supplementary Fig. 12b** and in **Lines 187-189**.

References:

Burkhard, P., *et al.* (1998). Three-dimensional structure of *O*-acetylserine sulfhydrylase from *Salmonella typhimurium*. *Journal of molecular biology*, 283(1), 121-133.

Bonner, E. R., *et al.* (2005). Molecular basis of cysteine biosynthesis in plants structural and functional analysis of *o*-acetylserine sulfhydrylase from *Arabidopsis thaliana*. *Journal of Biological Chemistry*, 280(46), 38803-38813.

Also, I'm wondering how the substitution to threonine show 50% of the activity to the wild-type protein. Did the authors analyze the phosphorylation of this OAS-TL with or without OAS? The different mobility in SDS-PAGE also suggested that this base substitution influenced the protein modification.

Response: We did not analyze the phosphorylation of OsASTOL1 with or without OAS, because we believe that the loss of OAS-TL activity of Osastol1 is unrelated to phosphorylation of Ser189 for the following reason: the astol1 mutation (Ser189Asn) is not a phosphomimic mutant. Among the different site-directed mutagenized variants of OsASTOL1, the S189 residue was replaced by either an alanine (A) to generate a constitutively nonphosphorylatable form, or with an aspartic acid (D) to mimic a constitutively phosphorylated form. The S189A protein variant and the S189D protein variant had no detectable OAS-TL activity *in vitro* (**Supplementary Fig. 14b**). This finding strongly suggests that OAS-TL activity of the ASTOL1 proteins is not affected by phosphorylation of Ser189 but that this point mutation affects the OAS binding as shown for the protein variant Ser189Asn (**Fig. 3d**). We have added the following sentences in the revised version of the manuscript to explain the above arguments: "Because both the mutation alleles of S189A (a constitutively nonphosphorylatable form) and S189D (a mimic of the constitutively phosphorylated form) lost the ASTOL1 activity completely, it is unlikely that phosphorylation of S189 is required for the enzyme activity." (**Lines 198-201**)

2. The authors need to demonstrate the role of this OAS-TL under the normal condition. The secondary structure of this isoform is similar to the cytosolic isoform of *Arabidopsis* but localized to the plastid. Is it the only OAS-TL existing in rice plastid?

Response: We provided direct evidence that OsASTOL1 has the cysteine synthase activity *in vivo* by complementation of the *E. coli* NK3 mutant (**Supplementary Fig. 12a**). Furthermore, we demonstrated OAS-TL activity of the ASTOL1 in *in vitro* enzyme assays (**Fig. 2e**). We also showed that knockout of OsASTOL1 by CRISPR/Cas9 did not affect growth or S metabolism (**Supplementary Figure 15**), which is in agreement with previous studies on different OAS-TL isoforms in *Arabidopsis* (Heeg *et al.*, 2008; Takahashi *et al.*, 2011). The rice genome encodes for three potential OAS-TL isoforms (**Supplementary Fig. 8**).

OsOAS-TL A is exclusively localized to the cytosol (Wang *et al.*, 2020). ASTOL1 is a plastid-localized OAS-TL protein in rice, as shown in our manuscript. The remaining Os1g0978100 might be a mitochondrial isoform, since in the model plant *Arabidopsis* OAS-TL activity is present in all compartments with own protein biogenesis (Heeg *et al.*, 2008; Watanabe *et al.*, 2008).

References:

Wang, C., *et al.* (2020). OsOASTL-A1 functions as a cytosolic cysteine synthase and affects arsenic tolerance in rice. *Journal of Experimental Botany*, 71, 3678–3689.

Heeg, C., *et al.* (2008). Analysis of the *Arabidopsis* O-acetylserine(thiol)lyase gene family demonstrates compartment-specific differences in the regulation of cysteine synthesis. *The Plant Cell*, 20, 168-185.

Takahashi, H., *et al.* (2011). Sulfur assimilation in photosynthetic organisms: molecular functions and regulations of transporters and assimilatory enzymes. *Annual Review of Plant Biology*, 62, 157-184.

Watanabe, M., *et al.* (2008). Physiological roles of the β -substituted alanine synthase gene family in *Arabidopsis*. *Plant physiology*, 146(1), 310-320.

In the case of *Arabidopsis*, there is a feedback inhibition for cytosolic OAS-TL activity, and the corresponding amino acid was demonstrated. Is the amino acid is conserved in this isoform?

Response: We are not aware of any report demonstrating feedback inhibition of cytosolic OAS-TL (cysteine synthase). All reports aiming at the feedback inhibition of cytosolic cysteine synthesis refer to the well-established feedback inhibition of SAT by cysteine (Noji *et al.* 1998; Kawashima *et al.* 2005). However, we might have missed the report mentioned by the reviewer and would like to ask the reviewer for the information of the report. Regardless, our results and previous studies on *Arabidopsis* (Heeg *et al.*, 2008) showed that a partial loss of OAS-TL activity did not impair growth or S metabolism. Loss-of-function of the OAS-TL activity in the mutant is irrelevant to the mutant phenotypes (Please see **Lines 205-218**). It is the altered SAT activity resulting from the stabilized CSC that produces the observed phenotypes of *astoll*.

References:

Heeg, C., *et al.* (2008). Analysis of the *Arabidopsis* O-acetylserine(thiol)lyase gene family

demonstrates compartment-specific differences in the regulation of cysteine synthesis. *The Plant Cell*, 20, 168-185.

Noji, M., *et al.* (1998). Isoform-dependent differences in feedback regulation and subcellular localization of serine acetyltransferase involved in cysteine biosynthesis from *Arabidopsis thaliana*. *Journal of Biological Chemistry*, 273(49), 32739-32745.

Kawashima, *et al.* (2005). Characterization and expression analysis of a serine acetyltransferase gene family involved in a key step of the sulfur assimilation pathway in *Arabidopsis*. *Plant Physiology*, 137(1), 220-230.

Because this is the semi-dominant mutation, there is a possibility to affect the activity of other OAS-TL by comprising the heterodimer if it is existing in the plastid.

Response: This conceivable hypothesis of the reviewer is based on a potential impact of the mutated *astol1* protein on other plastid localized OAS-TL isoform in rice. Hetero-dimerization between different OAS-TL isoforms have not been reported so far in plants, since the subcellular compartments of the so far analyzed plants species appear to possess only one true OAS-TL in each compartment (Takahashi, *et al.* 2011). Hetero-dimerization between different OAS-TL isoforms have also not been reported after expression of plant OAS-TLs in bacteria containing *cysK* (bacterial OAS-TL) and is unlikely to occur due to the large isoform-specific dimerization interface of OAS-TL proteins (Bonner, *et al.* 2005; Francois, *et al.* 2006). However, to address the reviewers concern by experimentation, we have mixed purified recombinant wild-type ASTOL1 and the mutated *astol1* protein in a 1:1 ratio and tested the impact of *astol1* on the ASTOL1 activity (**Please see the Figure below**). The addition of *astol1* did not impair the enzymatic activity of the wild-type ASTOL1 protein, suggesting that *astol1* does not form hetero-dimers with ASTOL1. We have not included these data in the revised manuscript because we believe they are peripheral to the main story.

Fig. *In vitro* enzyme activities of purified OsASTOL1, Osastol1 and their mixture.

In vitro OAS-TL enzyme activity of purified mature OsASTOL1 and Osastol1 and mixture of these two proteins. Data are shown as means \pm s.d. n = 3 technical replicates.

References:

Takahashi, H., *et al.* (2011). Sulfur assimilation in photosynthetic organisms: molecular functions and regulations of transporters and assimilatory enzymes. *Annual Review of Plant Biology*, 62, 157-184.

Bonner, E. R., *et al.* (2005). Molecular basis of cysteine biosynthesis in plants structural and functional analysis of *o*-acetylserine sulfhydrylase from *Arabidopsis thaliana*. *Journal of Biological Chemistry*, 280(46), 38803-38813.

Francois, J. A., *et al.* (2006). Structural basis for interaction of *O*-acetylserine sulfhydrylase and serine acetyltransferase in the Arabidopsis cysteine synthase complex. *The Plant Cell*, 18(12), 3647-3655.

Also, the native ASTOL1 seems localized to both plastid and cytosol comparing the mutant exactly localizing to the plastid in Fig 2 and Fig S10.

Response: With regard to localization, our data from transient expression in both rice and tobacco cells strongly suggest that OsASTOL1 is localized in the chloroplast (**Fig. 2a and Supplementary Fig. 10e**). Due to the strong promoter (35S) used in the transient expression, sometimes the strong YFP signal may overflow the chlorophyll fluorescence. However, the localization of the OsASTOL1-YFP signal is very different from that of the YFP alone. Also, OsASTOL1 has a typical signal peptide for chloroplast localization (**Table S2**).

3. The expression of this gene seems higher in the stem and sheath. As accumulation in the grain is determined by the chelation and sequestration around the node, authors need to

analyze the As and thiol levels in the other tissues than the grain or the seedlings.

Response: We have conducted a new experiment to determine thiol levels in the shoots and roots of 4-week-old plants, and the data are now presented in **Fig. 4f-g**. These data show that roots have much higher levels of PCs than shoots, and that the *astol1* mutant contains higher levels of PCs in both roots and shoots than WT. This new Figure replaces the original Figure containing only the thiol data in the roots.

We have also analyzed the As concentrations in the node I, internode, leaf blade and leaf sheath of the mutant and wild-type plants from two field experiments. The samples were collected previously at the plant maturity stage. The data are now presented in the new **Supplementary Fig. 19**, showing that As concentration in the node was increased in the mutant significantly at one field site, but not significantly at the other field site. Although rice nodes contribute to the sequestration of As (Moore *et al.*, 2014; Song *et al.*, 2014; Chen *et al.*, 2015), our previous experiment using radioactive ⁷³As isotope tracing showed that the majority (~90%) of As taken up by rice roots is sequestered in the roots (Zhao *et al.*, 2012). Therefore, lower grain As in the mutant is likely a result of increased As sequestration in the roots. We did not determine PCs in the nodes because the samples were taken previously at plant maturity from paddy fields and stored at room temperature, and PCs would probably have been partially degraded.

References:

- Moore, K. L., *et al.* (2014). Combined NanoSIMS and synchrotron X - ray fluorescence reveal distinct cellular and subcellular distribution patterns of trace elements in rice tissues. *New Phytologist*, 201(1), 104-115.
- Chen, Y., *et al.* (2015). The role of nodes in arsenic storage and distribution in rice. *Journal of Experimental Botany*, 66(13), 3717-3724.
- Song, W.-Y., *et al.* (2014). A rice ABC transporter, OsABCC1, reduces arsenic accumulation in the grain. *PNAS* 111, 15699-15704.
- Zhao, F. J., *et al.* (2012). Arsenic translocation in rice investigated using radioactive ⁷³As tracer. *Plant and Soil*, 350(1-2), 413-420.

4. Arsenite is known to be chelated by phytochelatin rather than GSH mostly in the node. The authors only presented the PC level in roots, but need to analyze PC levels in the grain and the different part of the plant.

Response: We agree that this would be a helpful information to understand the arsenite

tolerance of *astoll*. For that reason, we have performed a new experiment and determined the PCs in both roots and shoots of 4-week-old hydroponically grown plants that were treated with arsenite. These new results (**Fig. 4f-g, Lines 287-292**) demonstrate that arsenite induces PC synthesis in both organs. However, the accumulation of PCs was significantly higher in the roots of both genotypes when compared to shoots. In both organs, the *astoll*(+/-) mutant accumulated significantly more PCs than the wild type, strongly suggesting that the arsenite is predominantly chelated in the roots of *astoll*(+/-) and the wild type.

Understandably, sulfur accumulation in seedling is increased by the increased activity of sulfate transporter, but, strangely, seed S content in seeds is increased like this. I think the authors should discuss this point. Also, when the plastid OAS-TL is mutated, how the S assimilation can be increased? It is not a matter of sulfate uptake, but the matter of reduction and assimilation. I think the authors need to analyze the activity of whole OAS-TL activity in plants, if possible by separating the cytosol and plastid.

Response: We agree with the reviewer that not only sulfate uptake, but also its reduction and assimilation are important. The increased OAS not only enhances sulfate uptake, but also enhances sulfate reduction and assimilation (Nguyen *et al.*, 2012; Xiang *et al.*, 2018). As shown in **Fig. 4b**, in addition to sulfate, many sulfur-containing metabolites are also increased in the *astoll* (+/-) mutant, indicating that sulfate reduction and assimilation are enhanced. We have also added the expression data of adenosine phosphosulfate reductase *OsAPRI* and sulfite reductase *OsSiR* to **Fig. 4C**. The effect of OAS on both sulfate uptake and assimilation has been made clear in the Discussion (**Lines 362-364**). The reasons why S189N mutation in *Astoll1* can lead to enhanced sulfate uptake and assimilation are discussed fully in **Lines 347-362** of the Discussion section and in the model of **Fig. 6**. Several points are pertinent in the interpretations of our results here: 1) the OAS-TL activity is not the limiting factor for Cys biosynthesis, but rather the SAT activity is the limiting factor (Hell & Wirtz, 2011); 2) there is a functional redundancy between different isoforms of OAS-TL (Heeg *et al.*, 2008); 3) sulfide, OAS, and cysteine in the cytosol and different organelles can be exchanged efficiently (Heeg *et al.*, 2008; Takahashi *et al.*, 2011). These points and the fact that the SAT activity is enhanced in the mutant due to a more stable CSC complex fully explain the observed phenotypes of the mutant.

References:

Nguyen, H. C., *et al.* (2012). Improving the nutritive value of rice seeds: elevation of cysteine

and methionine contents in rice plants by ectopic expression of a bacterial serine acetyltransferase. *Journal of Experimental Botany*, 63(16), 5991-6001.

Xiang, X., *et al.* (2018). Overexpression of serine acetyltransferase in maize leaves increases seed-specific methionine-rich zeins. *Plant Biotechnology Journal*, 16(5), 1057-1067.

Heeg, C., *et al.* (2008). Analysis of the Arabidopsis O-acetylserine(thiol)lyase gene family demonstrates compartment-specific differences in the regulation of cysteine synthesis. *The Plant Cell*, 20(1), 168-185.

Takahashi, H., *et al.* (2011). Sulfur assimilation in photosynthetic organisms: molecular functions and regulations of transporters and assimilatory enzymes. *Annual Review of Plant Biology*, 62, 157-184.

Hell, R., & Wirtz, M. (2011). Molecular biology, biochemistry and cellular physiology of cysteine metabolism in Arabidopsis thaliana. *The Arabidopsis book* 9.

5. In Fig. 4, the authors show the ratio between the control condition and arsenite application, but the difference between the two genotypes is also important. Supplemental Table showing the metabolite levels would be requested. Also, I'm wondering why cystathionine content is so much affected rather than GSH.

Response: We thank the reviewer for this helpful comment to increase the readability of our manuscript and entirely agree to his/her suggestion. For that reason, **Fig. 4b** depicts the ratio of metabolites in the mutant to wild type and not the ratio between the control and +arsenite treatment in the wild type. In the revised version of the manuscript, we also included the exact metabolite levels in both genotypes to make this point clear and to allow for the evaluation of S-flux partitioning in the S-metabolism related network (**new Table S4**). In fact, the GSH steady-state level (7 nmol/g in WT) is three order of magnitude higher than the cystathionine steady-state level (0.002 nmol/g in WT) that the 20-fold increase in the value for cystathionine in the mutant does not add much to sulfur-flux partitioning in response to the *astol1* mutation. We hope that these additions will help readers to interpret the data presented.

Minor points

Fig 1g in L85 should be Fig 1h.

Response: Thank you for pointing this out. We have corrected this mistake in the revised version (**Line 95**).

The order of Supplemental Fig 8-10 should be changed to the order of their appearance in the

text.

Response: Thank you for pointing this out. We have changed this in the revised version.

Fig 4a is not cited in the text.

Response: Figure 4a is now cited in the revised version (**Line 265 and Line 267**).

Reviewer #3 (Remarks to the Author):

This is a very interesting work that identifies the molecular basis of the rice arsenic tolerance mutant *astol1*. Interestingly the mutation confers an increased stability of the Cysteine Synthase Complex (CSC) due to a point mutation on the OASTL rice protein. The mutation also affects OASTL enzyme activity leading to increased OAS accumulation. Based on the important regulatory role of OAS in cysteine biosynthesis the mutant displays increased sulphur, glutathione and phytochelatin content that leads to increased arsenic accumulation in roots preventing its accumulation in grains. In addition, the mutant also increased Selenium accumulation. Therefore, uncovering the molecular mechanisms underlying *astol1* mutation, provides a novel strategy to obtain plants with less accumulation of arsenic in the aerial part together with increased selenium accumulation.

Response: We thank the reviewer for the very positive comments.

I believe the quality of the results obtained are excellent, however I would like to add some comments regarding the way in which the work is introduced and described.

I think the CSC complex and functionality should be further explained in the introduction. Previous work regarding this issue should be mentioned. Indeed, it has been shown that 80% of the SAT activity comes from the mitochondrial isoform while OASTL activity comes preferentially from the plastidial and cytosolic isoforms. Therefore, the role of CSC in each compartment is currently controversial and that should be further discussed in the context of the obtained data.

Response: We agree with the reviewer and included more information about the regulatory function of the CSC in the introduction (**Lines 49-64**). Furthermore, we have added the following sentences in the discussion (**Lines 392-402**) to dissect the role of the CSC in the subcellular compartments of the plant cell for regulation of cellular cysteine synthesis rate: ...“Remarkably, these sub-cellular compartments are also equipped with a CSC, which is

supposed to regulate the OAS precursor biogenesis by controlling the SAT activity in these compartments. Since 80% of total SAT activity is localized in mitochondria (Ruffet et al., 1995; Watanabe et al., 2008), the mitochondrial CSC is supposed to efficiently regulate cellular OAS synthesis in plants (Wirtz et al., 2010). Accordingly, downregulation of mitochondrial SERAT2;1 impairs growth of *Arabidopsis* (Haas et al., 2008). However, expression of SAT in the cytosol of *Arabidopsis* and *Nicotiana tabacum*, also resulted in enhanced cysteine formation, demonstrating that enhanced OAS production in these compartments also trigger net cysteine synthesis rate (Noji & Saito, 2002; Wirtz & Hell, 2007). In particular, the plastid localized CSC is supposed to integrate environmental cues to coordinate cysteine biosynthesis for GSH production to maintain redox homeostasis upon diverse stresses (Dominguez-Solis et al., 2008; Park et al., 2013; Mueller et al., 2017). In contrast to OAS biosynthesis...”

In addition, both OASTL and SAT genes are not transcriptionally regulated but exposure to stress conditions can induce transcript accumulation (Lehmann et al 2009. Mol Plant 2:390-406). Therefore, quantification of transcript abundance in response to As(III) in wild-type and mutant plants would be desirable and the results discussed under the context of previous data.

Response: Thank you for the suggestion. To address the reviewer’s concern, we have conducted a new experiment to determine the expression levels of *OAS-TLs* and *SATs* genes in WT and *astol1(+/-)* in response to arsenite exposure. The new data are presented in **Supplementary Fig. 17a-d**, which shows no significant differences in the expression levels of those genes between WT and *astol1(+/-)* when treated with As(III) (**Lines 289-292**).

The importance of arsenic in rice natural variation should be explored. I would suggest to look for the *astol1* SNP or analogous in the sequence database of different rice accessions. It would be particularly relevant if the arsenic tolerance of the accessions were already determined, which would further support the biological importance of the mutation uncovered here and therefore improve the quality of the results presented.

Response: Thank you for the comment. We have searched the *astol1* SNP site of *ASTOL1* in 4,726 rice accessions at RiceVarMap database (<http://ricevarmap.ncpgr.cn/v2/>), but have not found any natural variation in this site. This is not surprising as the S189 residue is highly conserved.

Finally, I would further discuss the possible reasons that lead to the pleiotropic effects of the mutation, in particular the effect of light intensity.

Response: We followed the reviewer's suggestion and added the following sentences in the discussion section to address the possible reasons for the light-dependent differences in growth: "Increased biosynthesis of OAS, Cys and other elevated thiol compounds in the *astol1* mutants causes a higher demand for carbon. Consequently, norm light conditions helped the mutants to overcome the imposed challenge for enhanced synthesis of the diverse carbon and sulfur-containing metabolites. The relatively poor growth of the mutants under low light intensity in the absence of any stresses might be also partially explained by a misbalance of enhanced level of sulfur-containing defense compounds in the artificial absence of stresses, e.g., high light and pathogens, which demand sulfur-containing defense compounds. However, in open paddy fields the potentially defense-primed *astol1*(+/-) mutant performed as well as the wild type, since the wild type was also challenged with environmental cues inducing sulfur-defense metabolite synthesis." (**Lines 362-374**).

In conclusion, I think this is an excellent work providing a new perspective to obtain rice plants with better performance in arsenic contaminated soils preventing its accumulation in grains and increased selenium resulting in fortified crops. The results are novel, the conclusions are very well sustained and the methods are sufficiently explained.

Response: Thank you again for the positive comments about our work.

Minor comments:

- Figure 1g cited in line 85 should be Figure 1h.

Response: Thank you for pointing this out. We have corrected this in the revised version (**Line 95**).

- In line 150 it must be said that *astol1* point mutation does not affect the OASTL chloroplastic localization in rice

Response: Thank you for pointing this out. We have corrected this in the revised version (**Line 161**).

- The complementation of NK3 complementation assay should be explained with further detail.

Response: Thank you for the comment. Please see **Lines 177-189** in the revised version.

- The use of the OASTL cytosolic isoform from Arabidopsis should be further justified.

Response: Thank you for the comment. Please see **Lines 174-177** in the revised version.

REVIEWERS' COMMENTS

Reviewer #1 (Remarks to the Author):

Because all of my concerns have been properly revised, I have no issues with the present manuscript being accepted.

Satoru Ishikawa, NARO Japan

Reviewer #2 (Remarks to the Author):

I think the authors addressed well for all the comments made in the first review. Now it seems no issues to be raised by me.

Reviewer #3 (Remarks to the Author):

I think the authors made a good effort responding to the Reviewer's criticism and comments. There is no doubt that the new experiments included in this version improved the quality of the work.